# What Can RL Bring to VLA Generalization? An Empirical Study

**Jijia Liu**[1]* **Feng Gao**[2]* **Bingwen Wei**[1]
**Xinlei Chen**[1] **Qingmin Liao**[1] **Yi Wu**[2] **Chao Yu**[1]† **Yu Wang**[3]†

[1]Shenzhen International Graduate School, Tsinghua University
[2]Institute for Interdisciplinary Information Sciences, Tsinghua University
[3]Department of Electronic Engineering, Tsinghua University

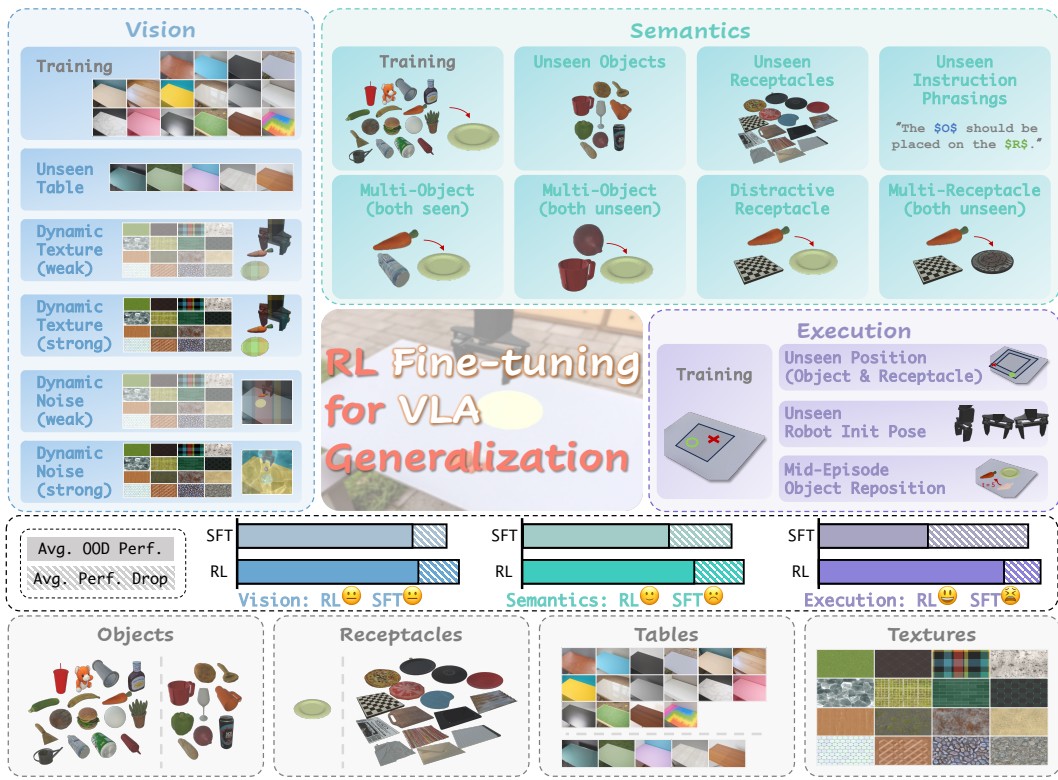

Figure 1: Overview of our study for evaluating how RL enhances VLA generalization in terms of *Vision*, *Semantics*, and *Execution*: in out-of-distribution tests, RL yields substantial gains in *Execution*, moderate improvements in *Semantics*, and performance on par with SFT for *Vision*.

## Abstract

Large Vision-Language Action (VLA) models have shown significant potential for embodied AI. However, their predominant training via supervised fine-tuning (SFT) limits generalization due to susceptibility to compounding errors under distribution shifts. Reinforcement learning (RL) offers a path to overcome these limitations by optimizing for task objectives via trial-and-error, yet a systematic understanding of its specific generalization benefits for VLAs compared to SFT is lacking. To address this, our study introduces a comprehensive benchmark for evaluating VLA generalization and systematically investigates the impact of

*Equal contribution.
†Corresponding authors: {yuchao, yu-wang}@mail.tsinghua.edu.cn

39th Conference on Neural Information Processing Systems (NeurIPS 2025).

RL fine-tuning across diverse visual, semantic, and execution dimensions. Our extensive experiments reveal that RL fine-tuning, particularly with PPO, significantly enhances generalization in semantic understanding and execution robustness over SFT, while maintaining comparable visual robustness. We identify PPO as a more effective RL algorithm for VLAs than LLM-derived methods like DPO and GRPO. We also develop a simple recipe for efficient PPO training on VLAs, and demonstrate its practical utility for improving VLA generalization. The project page is at `https://rlvla.github.io`.

# 1 Introduction

Vision-Language-Action (VLA) models represent an emerging class of foundation models [Ma et al., 2024, Firoozi et al., 2023] that unify perception, language understanding, and embodied control. By leveraging vision-language models pretrained on internet-scale data and further training on large, heterogeneous robot demonstration datasets [Collaboration et al., 2023, Khazatsky et al., 2024], VLAs can interpret sensor observations and natural language instructions to directly map them to robot actions. This paradigm has demonstrated promising generalization across diverse tasks—including single-arm, bimanual, and mobile manipulation [Team et al., 2024, Kim et al., 2024, Liu et al., 2024, Wen et al., 2025], navigation [Shah et al., 2022], and even complex long-horizon activities like kitchen or bedroom cleaning in unseen scenarios [Black et al., 2024, Intelligence et al., 2025].

Despite this promise, VLA model training predominantly relies on supervised fine-tuning (SFT) through behavioral cloning of demonstration labels [Kim et al., 2025]. This approach, whether in pretraining or few-shot adaptation, is inherently susceptible to compounding errors under distribution shift: minor deviations from expert trajectories can accumulate, steering the policy into unfamiliar states and undermining robust performance [Ross and Bagnell, 2010, De Haan et al., 2019, Belkhale et al., 2023, Foster et al., 2024]. This mismatch between training and testing distributions fundamentally limits robustness, with research highlighting issues like quadratic regret growth relative to the task horizon under such conditions [Ross and Bagnell, 2010].

In contrast, reinforcement learning (RL) offers a paradigm that directly optimizes cumulative task rewards through trial-and-error, enabling policies to explore beyond narrow expert data and learn corrective behaviors. Crucially, in the broader foundation model landscape, particularly for Large Language Models (LLMs) and Vision-Language Models (VLMs), recent studies have underscored RL's advantages for generalization [Ouyang et al., 2022, Zhai et al., 2024, Huang et al., 2025]. Compelling evidence suggests that while SFT tends to memorize training data, RL fine-tuning can lead to substantially better out-of-distribution performance and unlock greater reasoning capabilities [Chu et al., 2025, Huang et al., 2025, Ma et al., 2025]. Drawing from RL's established success in robotics and these encouraging results from other large-scale models, RL fine-tuning is increasingly being applied to VLAs [Collaboration et al., 2023, Walke et al., 2023, Khazatsky et al., 2024], with approaches ranging from incorporating human feedback [Chen et al., 2025] to using offline RL updates [Zhang et al., 2024b] or algorithms like PPO [Schulman et al., 2017], sometimes in multi-stage processes with imitation learning [Guo et al., 2025b].

However, despite these pioneering efforts, a systematic understanding of what specific generalization benefits RL fine-tuning confers upon VLAs, especially in direct comparison to SFT baselines, and how their respective strengths differ, remains insufficiently developed [Hu et al., 2024, Mark et al., 2024]. For instance, while recent work like FLaRe [Hu et al., 2024] demonstrated PPO's utility for fine-tuning VLAs, a comprehensive analysis of the resulting model's generalization capabilities was not its primary focus.

This paper aims to address this critical gap. We undertake a systematic study to dissect the generalization properties of VLAs fine-tuned with RL versus SFT.

> *Specifically, we empirically investigate: What unique benefits can RL bring to VLA generalization compared to supervised fine-tuning?*

To answer this, we center our investigations on the representative *pick-and-place* task. On this task, we first evaluate mainstream RL algorithms for large-scale models (PPO [Schulman et al., 2017, Ouyang et al., 2022], DPO [Rafailov et al., 2023, Zhang et al., 2024b], GRPO [Shao et al., 2024, Guo et al., 2025a]) to identify effective VLA fine-tuning strategies. We then conduct a broad

comparative evaluation, also on pick-and-place, pinpointing where RL fine-tuning outperforms SFT. As previewed in Fig. 1, this comprehensive evaluation rigorously examines generalization across three key dimensions: 1) *Vision*: challenging generalization with novel backgrounds (unseen tables) and by overlaying unseen textures on the foreground or the entire image. 2) *Semantics*: testing understanding via unseen objects, novel receptacles, and varied instruction phrasings to probe language sensitivity and object recognition. 3) *Execution*: probing robustness by varying initial robot states, object/receptacle positions, and introducing dynamic disturbances like random object reposition within episodes.

Through extensive experiments and deep analyses, we:

- Establish a rigorous and challenging benchmark to evaluate how VLA fine-tuning methods affect generalization across diverse visual, semantic, and execution dimensions.
- Identify PPO as the preferred RL algorithm for VLA fine-tuning over GRPO and DPO, while also discuss key challenges in adapting these RL algorithms from LLM/VLM paradigms to the distinct requirements of VLAs.
- Develop an efficient PPO-based VLA fine-tuning recipe, enabled by a shared actor-critic backbone, VLA model warm-up, and minimal PPO training epochs.
- Demonstrate RL's superior generalization over SFT in semantic understanding and embodied execution for VLAs, while maintaining comparable visual robustness.

## 2 Related works

### 2.1 Vision-Language-Action (VLA) models

In robotic tasks that utilize both vision and language inputs, recent research demonstrates that leveraging knowledge from pre-trained LLMs or VLMs [Chen et al., 2023, Driess et al., 2023, Karamcheti et al., 2024, Beyer et al., 2024] leads to better policy generalization compared to smaller models [Zhao et al., 2023, Chi et al., 2023]. By harnessing the capabilities of these foundation models and utilizing large-scale robotic datasets, VLA models [Brohan et al., 2023b,a, Team et al., 2024, Kim et al., 2024, Liu et al., 2024, Wu et al., 2023, Cheang et al., 2024, Black et al., 2024, Intelligence et al., 2025] provide a strong basis for embodied agent. Our work builds upon VLA and seeks to further improve its generalization abilities.

### 2.2 RL fine-tuning for large-scale (Vision-)Language Models

Fine-tuning large language models has shown promising results [Stiennon et al., 2020, Ouyang et al., 2022, Achiam et al., 2023], especially when using high-quality supervised data [Zhang et al., 2024a, Zhu et al., 2023, Zhang et al., 2023b]. Some work aims to align LLMs with human preferences for helpfulness and safety [Kaufmann et al., 2023, Bai et al., 2022], using online RL algorithms [Schulman et al., 2017, Ramamurthy et al., 2023, Zheng et al., 2023] or offline updates [Rafailov et al., 2023, Ivison et al., 2024, Xu et al., 2024]. Other studies have shown that online RL fine-tuning on math and coding tasks [Shao et al., 2024, Guo et al., 2025a] can significantly improve the reasoning abilities of LLMs [Guo et al., 2025a, Jaech et al., 2024, Team et al., 2025]. Although RL has been extensively explored in LLMs [Chu et al., 2025], its generalization potential for similar-scale VLA models remains underexplored.

### 2.3 Fine-tuning policies for robotic tasks

VLA fine-tuning largely relies on supervised data [Kim et al., 2025, Liu et al., 2025, Zhao et al., 2025, Zheng et al., 2024]. However, the scarcity of extensive and high-quality data often hampers the ability of VLAs to generalize to unseen scenarios or perturbations [Collaboration et al., 2023, Walke et al., 2023, Khazatsky et al., 2024]. To move beyond imitation and enable reward-driven learning, RL fine-tuning has been explored in various settings, from small-scale models [Ren et al., 2024] to large models with human interventions [Chen et al., 2025], or through offline updates [Zhang et al., 2024b]. Guo et al. [2025b] proposed using PPO [Schulman et al., 2017] to fine-tune VLA models, incorporating imitation learning in a two-stage process. Despite these efforts, the generalization ability of RL fine-tuning in VLA models remains insufficiently studied [Hu et al., 2024, Mark et al., 2024]. Moreover, we find that VLAs can be directly fine-tuned with online PPO without additional training stages or imitation learning, underscoring RL's scalability for robot tasks.

# 3 Preliminaries

## 3.1 Problem formulation

We model each language-conditioned robotic task $T \in \mathcal{T}$ as a Partially Observable Markov Decision Process (POMDP), defined as a tuple $\mathcal{M} = (\mathcal{S}, \mathcal{A}, \mathcal{P}, R, \mathcal{O}, \mathcal{L}, P(s_0), \gamma)$. Here, the state space $\mathcal{S}$ includes robot and environment states, $\mathcal{A}$ is the action space of control commands, and $\mathcal{O}$ the observation space of sensor outputs. Transitions follow $s_{t+1} \sim \mathcal{P}(\cdot \mid s_t, a_t)$, and $\gamma$ is the discount factor. Each task $T$ has natural language instructions $\mathcal{L}_T \subset \mathcal{L}$ and a reward $R(s, l) \in \mathbb{R}$ when state $s$ completes task stages or satisfies instruction $l$. Episodes start with an instruction $l \sim \mathcal{L}_T$ and initial state $s_0 \sim P(s_0)$, which defines the robot's initial pose and environment. The policy $\pi_\theta$ uses the last $H$ observations $o_{t-H+1:t}$ and the language instruction $l$ to output actions $a_t \sim \pi_\theta(a_t \mid o_{t-H+1:t}, l)$, constructing trajectory $\tau = (o_0, a_0, \dots)$.

**Supervised fine-tuning (SFT).** SFT learns from an expert-collected demonstration dataset $\mathcal{D}_T = \{(\boldsymbol{\tau}^{(i)}, l^{(i)})\}_{i=1}^N$, where each trajectory is $\boldsymbol{\tau}^{(i)} = (o_0^{(i)}, a_0^{(i)}, \dots, o_{K_i-1}^{(i)}, a_{K_i-1}^{(i)})$ and N is the total number of trajectories. The VLA policy $\pi_\theta$ minimizes:

$$\mathcal{L}_{\text{SFT}}(\theta) = \sum_{(\tau^{(i)}, l^{(i)}) \in \mathcal{D}_T} \sum_{t=0}^{K_i-1} \ell_{\text{SFT}}(\hat{a}_t^{(i)}, a_t^{(i)}), \quad \hat{a}_t^{(i)} = \pi_\theta(o_{t-H+1:t}^{(i)}, l^{(i)}),$$

where $\ell_{\text{SFT}}$ is a loss function (e.g., next-token prediction, $L_1$ regression, diffusion) based on VLA architecture and action representation [Kim et al., 2025].

**RL fine-tuning.** RL fine-tuning maximizes scalar rewards or preference signals through direct interactions with environments. With rewards, it minimizes the negative discounted return over an episode of length $M$:

$$\mathcal{L}_{\text{RL}}(\theta) = -\mathbb{E}_{\boldsymbol{\tau} \sim \pi_\theta}\Big[\sum_{t=0}^{M-1} \gamma^t R(s_t, l)\Big],$$

often via a policy-gradient surrogate:

$$\mathcal{L}_{\text{PG}}(\theta) = -\mathbb{E}_{\tau \sim \pi_\theta}\Big[\sum_{t=0}^{M-1} A_t^\pi \log \pi_\theta(a_t \mid o_{t-H+1:t}, l)\Big],$$

using an advantage estimator $A_t^\pi$. With preferences, methods like Direct Preference Optimisation (DPO) [Rafailov et al., 2023] use pairwise/ranked feedback to optimize $\pi_\theta$ for preferred trajectories.

## 3.2 Vision-Language-Action models

We base our study on OpenVLA [Kim et al., 2024] (Fig. 2), an open-source model that achieves state-of-the-art performance by pairing a fused visual encoder of SigLIP [Zhai et al., 2023] and DI-NOv2 [Oquab et al., 2023] with a Llama-2 7B language backbone [Touvron et al., 2023], built on the Prismatic VLM [Karamcheti et al., 2024]. At each time step the policy receives a single RGB image $o_t$ and an instruction $l$, i.e., the history length $H = 1$. The image is embedded into visual tokens, the instruction is tokenised with Llama 2's

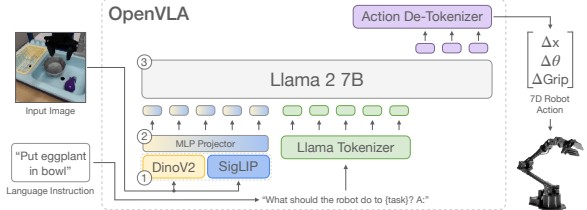

Figure 2: Architecture of the OpenVLA model [Kim et al., 2024], reproduced from the official open-source code and checkpoints.

tokenizer, and the resulting token sequence is fed to the causal transformer decoder.

OpenVLA follows the RT-2 discretisation recipe [Brohan et al., 2023a]: each scalar in the continuous command $a_t \in \mathbb{R}^{d_a}$ is mapped to one of 256 bins that uniformly split the range between its 1st and 99th percentiles in the training set, producing an action–token vector $\mathbf{u}_t \in \{0, \dots, 255\}^{d_a}$. These action tokens overwrite the 256 least-used tokens in the Llama-2 vocabulary, so the language model can emit them directly. The network is then trained with the usual next-token cross-entropy objective, and the cross-entropy loss is computed solely on the predicted action tokens.

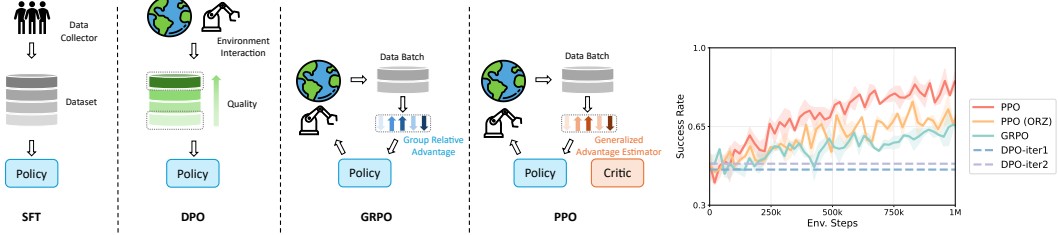

(a) Overview of different fine-tuning methods      (b) Performance comparison

Figure 3: Overview of VLA fine-tuning methods: SFT learns from offline demonstrations, whereas DPO, GRPO, and PPO use RL-based updates—employing preference alignment, group-relative advantage estimation, and standard actor-critic PPO with generalized advantage estimation (GAE), respectively; and performance comparison between different RL fine-tuning algorithms.

# 4 Effective RL fine-tuning of VLA models

We begin our study by exploring how to fine-tune VLA models with reinforcement learning. As these models inherit a similar scale and structure from their pre-trained LLM/VLM backbones, we first test whether common RL algorithms for large-scale models like LLMs/VLMs transfer effectively to VLAs. In Sec. 4.1, we will show that Proximal Policy Optimization (PPO) [Schulman et al., 2017] consistently improves performance, whereas Direct Preference Optimization (DPO) [Rafailov et al., 2023, Zhang et al., 2024b] and Group Relative Policy Optimization (GRPO) [Shao et al., 2024] struggle to learn something for POMDP robotic tasks. Guided by this finding, we then ablate key design choices within PPO to distill an effective fine-tuning recipe for VLA models in Sec. 4.2.

## 4.1 RL algorithms: PPO, GRPO, DPO

Here, we consider three representative RL algorithms: PPO [Schulman et al., 2017], GRPO [Shao et al., 2024] and DPO [Rafailov et al., 2023, Zhang et al., 2024b]. All models are fine-tuned using Low-Rank Adaptation (LoRA) [Hu et al., 2022] with rank = 32.

- **PPO** performs on-policy updates, computing policy gradients with clipped importance ratios and Generalised Advantage Estimation (GAE) from freshly collected rollouts. Recent work Open-Reasoner-Zero (ORZ) by [Hu et al., 2025] shows that disabling GAE (setting $\gamma = 1$ and $\lambda = 1$) can boost performance when fine-tuning large models. Accordingly, we evaluate both the standard algorithm (**PPO**) and this variant (**PPO-ORZ**) in our experiments.

- **GRPO** estimates the baseline using a group of samples, allowing for direct computation of advantages without explicit value estimation. While natural language generation and robotic multi-step MDP tasks are inherently different, we follow the GRPO setup [Shao et al., 2024] by sampling trajectories from a common initial state, with each group containing 8 trajectories.

- **DPO** exploits offline datasets annotated with pairwise preferences. In robotic tasks, however, obtaining contrastive trajectories from the same initial state is difficult. Following [Zhang et al., 2024b], we infer trajectory-level preferences from reward signals. To ensure a fair comparison, we employ only the stage-based sparse reward described in Sec. 5.1, rather than the richer reward scheme used in the original implementation by [Zhang et al., 2024b].

These algorithms are illustrated in Fig. 3a, with further details and implementation specifics provided in Sec. A.1. We evaluate the performance on the *pick-and-place* task described in Sec. 5.1.

The results are presented in Fig. 3b, with each experiment conducted using two different random seeds. Our findings indicate that PPO consistently outperforms GRPO. We attribute this to fundamental differences in environment dynamics between natural language tasks and robotic tasks [Li et al., 2024b]. We hypothesis that, in robotic tasks' POMDP, each action sequentially alters the environment state in a non-stationary manner, which may destabilizes GRPO's advantage estimates. Additionally, PPO surpasses DPO performance. We hypothesize that this is due to the sparse reward structure, which makes it challenging to distinguish between the quality of different trajectories, as well as significant distribution shifts between the offline dataset and interactive execution [Prudencio et al., 2023].

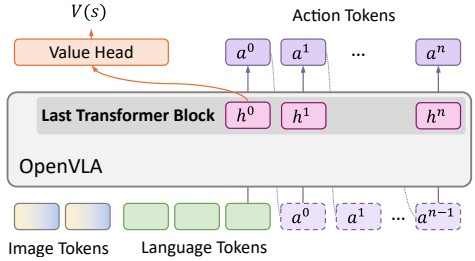
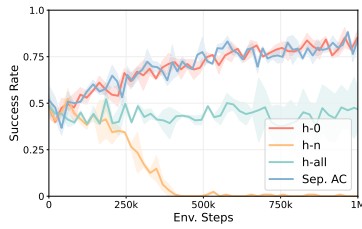

(a) Model architecture of actor-critic OpenVLA       (b) Performance comparison

Figure 4: PPO with shared actor-critic backbone, where $V(s)$ is predicted by a three-layer MLP. We compare the performance of different critic designs, as well as a separate actor-critic architecture.

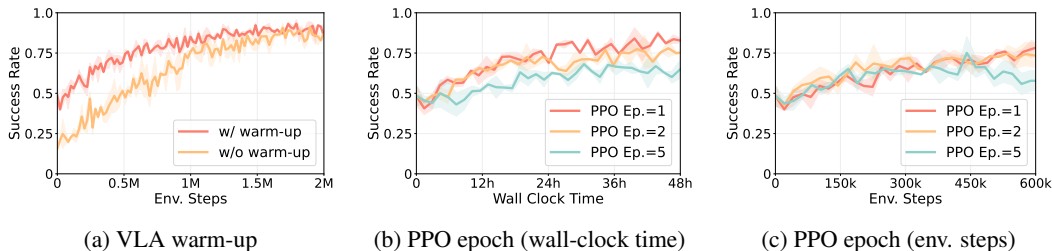

(a) VLA warm-up       (b) PPO epoch (wall-clock time)       (c) PPO epoch (env. steps)

Figure 5: Performance comparison to verify our efficient training designs.

## 4.2 Design factors of PPO

We employ several key design choices to enable PPO to work effectively with OpenVLA. With these designs, our main experiments require about 42 hours on a single NVIDIA A100 GPU to converge. In the following, we introduce these design choices and present ablation studies to assess their impact. We conduct each experiment with two different random seeds.

**Shared actor-critic backbone.** Since PPO—and most modern RL methods—use an actor–critic formulation [Konda and Tsitsiklis, 1999], we treat the pretrained VLA policy as the *actor* and attach a lightweight *critic* that estimates the state value $V(s)$. To keep the architecture compact, the actor and critic share the entire Transformer backbone; a three-layer MLP value head (see Fig. 4a) takes the hidden vector $h^0$ produced by the final Transformer block [Vaswani et al., 2017] at the first action-token position and regresses it to a scalar value. To verify its efficacy, we evaluated several critic designs. Feeding the value head with the first action-token embedding $h^0$ produced the highest and most stable returns, outperforming the last-token input $h^n$ and the concatenation $[h^0, \ldots, h^n]$ (see Fig. 4b). A critic that uses its own Transformer backbone achieved comparable rewards but trained $35\%$ slower and consumed $83\%$ more VRAM ($81.3$ GB vs. $44.4$ GB). These results confirm that a shared backbone with $h^0$ is the most efficient choice.

**VLA warm-up.** We run all experiments with the official OpenVLA checkpoint [Kim et al., 2024] pretrained on the OXE dataset [Collaboration et al., 2023]. Due to the poor performance of this checkpoint on our benchmark, we *warm it up* with $140$ demonstration trajectories gathered by Octo-Small [Team et al., 2024] and a motion planner (see Sec. A.2). As shown in Fig. 5a, the warm-up model reaches convergence with roughly $50\%$ fewer environment steps, while both initialisations attain comparable asymptotic returns given sufficient interaction. Unless stated otherwise, all subsequent experiments use the warmed-up OpenVLA model for initialization.

**Minimal PPO epoch.** We show that the update–to–data ratio is a key factor for efficient fine-tuning. In PPO this ratio is set by the *epoch* hyper-parameter, which specifies how many gradient passes each batch receives. Fig. 5b and Fig. 5c reveal that increasing the PPO *epoch* beyond one yields no gains in return or sample efficiency, yet lengthens wall-clock time almost linearly. Accordingly, we fix epoch $= 1$ in all remaining experiments, achieving the fastest training without sacrificing performance.

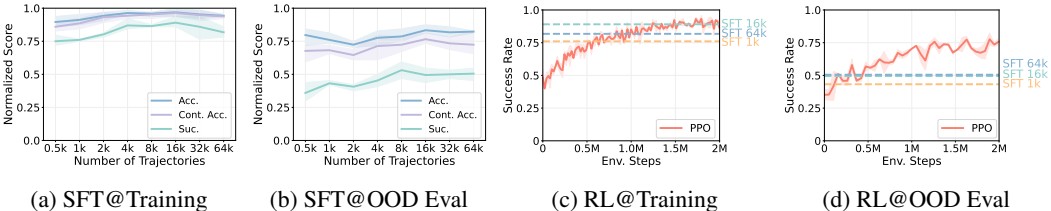

| (a) SFT@Training | (b) SFT@OOD Eval | (c) RL@Training | (d) RL@OOD Eval |

Figure 6: Effect of data scale on SFT performance. The reported metrics include grasp accuracy, continuous grasp accuracy, and task success rate.

# 5 Evaluating fine-tuning methods on VLA generalization

## 5.1 Environments and datasets

To thoroughly investigate the generalization capabilities of VLA, we focus on a typical *pick-and-place* task [Kim et al., 2025, Li et al., 2024a], where the agent is instructed to place an object from the table into a receptacle. Inspired by prior works [Fan et al., 2025, Stone et al., 2023] and the concept of Vision-Language-Action models, we define three dimensions of generalization: *Vision*, *Semantics (Language)*, and *Execution (Action)*, as illustrated in Fig. 1.

- *Vision*: We include both foreground (*Dynamic Textures*) and background (*Unseen Table*) changes, as well as image-level *Dynamic Noise*, applied with either *weak* or *strong* intensity.

- *Semantics*: We consider previously unseen variations in *Objects*, *Receptacles*, and *Instruction Phrasings*. Additionally, we design new tasks where one of two objects must be picked (*Multi-Object*), using either seen or unseen object sets. We also include a task with a *Distractive Receptacle*, and a task requiring the object to be placed in one of two unseen receptacles (*Multi-Receptacle*).

- *Execution*: We investigate changes in the initial positions of both the object and the receptacle, as well as the robot initial pose. Furthermore, we introduce a new scenario in which the object's position changes during the task (*Mid-Episode Object Reposition*).

To probe generalisation, we randomise each task along three axes during training: *Vision* (16 tables), *Semantics* (16 objects), and *Execution* (perturbations of object and receptacle poses). At test time we hold at least one of these factors out of distribution, introducing nine novel objects, sixteen unseen receptacles, five new table surroudings, and sixteen distractor textures. Assets are drawn from Objaverse [Deitke et al., 2023] and other public sources; additional table appearances are synthesised with Stable Diffusion [Rombach et al., 2022] and ControlNet [Zhang et al., 2023a], ensuring surface variation without changing the table's global pose. More details of asset acquisition and task specification are provided in Secs. A.3 and A.4. Built upon these assets, our tasks run in ManiSkill [Tao et al., 2024] with an 8-DoF WidowX-250S arm. At every step the agent observes a $640 \times 480$ RGB frame and a natural-language instruction, and outputs a Cartesian end-effector delta plus a binary gripper signal. Rewards are sparse: $0.1$ for grasping and continuously holding the correct object, and $1.0$ for placing it successfully. For supervised fine-tuning we collect demonstration trajectories using the MPLib motion planner [Guo et al., 2024] and fine-tuned using LoRA [Hu et al., 2022]; further dataset details can be found in Sec. A.2.

## 5.2 Effect of data scale on SFT performance

Following Lin et al. [2025], we study how supervised fine-tuning (SFT) scales with demonstration count. We train OpenVLA to convergence on datasets ranging from a few hundred to 64k expert trajectories ($\sim 1.26$M transitions) and report average scores over three random seeds in-distribution and on unseen objects/tables (Fig. 6a and 6b). We observe that performance plateaus at roughly 16k trajectories in both settings, so we adopt the 16k-trajectory SFT checkpoint as the baseline for comparing RL fine-tuning methods.

## 5.3 Performance comparison between RL and SFT

Building on preceding analyses, we plot success rates during RL training for both the training distribution and the OOD object/table split in Fig. 6c and 6d, alongside SFT scores at various data

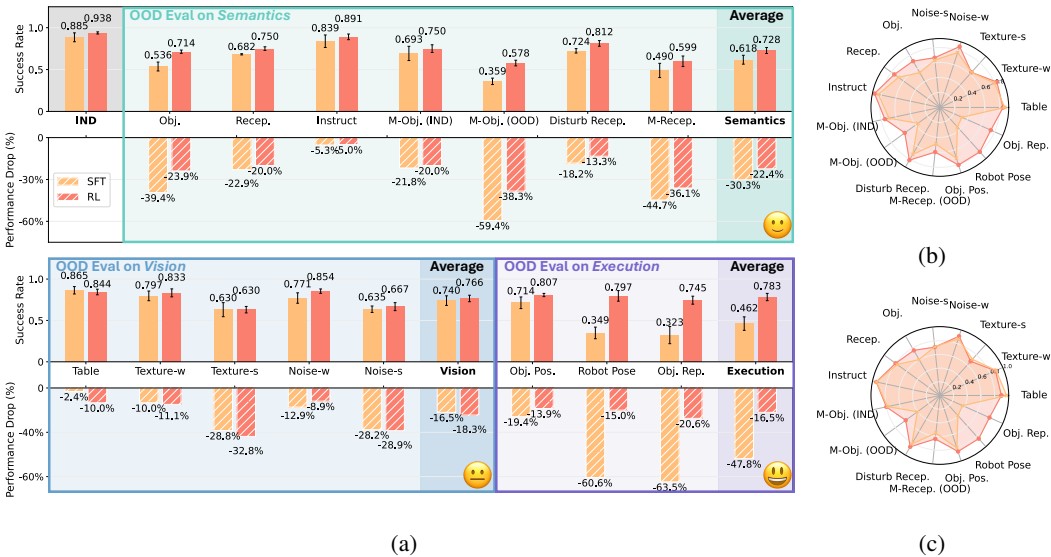

(a)                 (c)

Figure 7: (a) Performance comparison between SFT and RL across different tasks. Both success rate and relative performance drop are reported. (b) Radar chart of success rate. (c) Radar chart of the performance ratio of out-of-distribution (OOD) to in-distribution (IND).

scales. RL overtakes the strongest SFT checkpoint (SFT-16k) after roughly $0.4$ M environment steps on OOD task. At convergence, it performs comparably to SFT-16k in the training setting and $42.6\%$ better on unseen objects and tables, showing that reinforcement learning not only improves policy performance under the training distribution but also provides markedly stronger generalization.

We further evaluate the generalization performance of the converged RL policy and the best-performing SFT policy (SFT-16k, shortly denoted as SFT) on OOD tasks, as shown in Fig. 7. Here we report average success rates and average performance drop over three random seeds while full detailed results can be found in Sec. B.3. The relative performance drop is calculated as $P = \frac{\text{OOD}-\text{IND}}{\text{IND}}$. The results indicate that RL performs comparably to SFT in *Vision* tasks, noticeably better in *Semantics*, and significantly better in *Execution*.

For *Vision*, we hypothesize that neither RL nor SFT training induces visual robustness beyond the imposed visual randomness (i.e., random training tables), resulting in similar performance for both methods. In *Semantics* tasks, RL notably outperforms SFT when faced with OOD objects, both in single and multiple object scenarios. We hypothesize that, through trial-and-error, RL is able to learn the skill of "grasp" in a manner that is less dependent on object type, leading to improved generalization. For *Execution* tasks, RL surpasses SFT across all three evaluated scenarios: OOD object and receptacle positions, OOD robot initial positions, and mid-episode object repositioning.

### 5.4 Dissecting RL's contributions to generalization

To probe qualitative differences between the two approaches, we visualise policy roll-outs on four representative tasks (Fig. 9). In *Vision/Dynamic-Noise (strong)*, the SFT agent repeatedly drops the object immediately after grasping, indicating that heavy visual perturbations prevent it from localising the receptacle, whereas the RL agent completes the placement. With an *Unseen Object* in *Semantics*, SFT keeps attempting to grasp the item it already holds and stalls, while RL lifts it and finishes the task. This may be attributed to RL's extensive trial-and-error experience during training, enabling it to make finer-grained control when interacting with unseen objects. RL also learned to recover from failed grasps and mid-episode

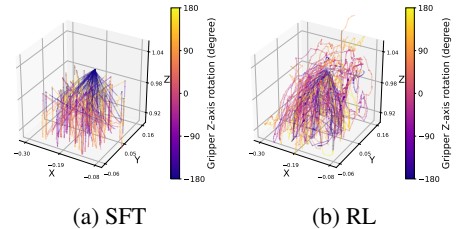

(a) SFT       (b) RL

Figure 8: Training trajectories of SFT (left) and RL (right); color encodes gripper Z-axis rotation.

object shifts in two *Execution* tests, whereas SFT marches on in spite of position errors, likely because such cases never appear in the demonstration data. The trajectory distributions encountered during

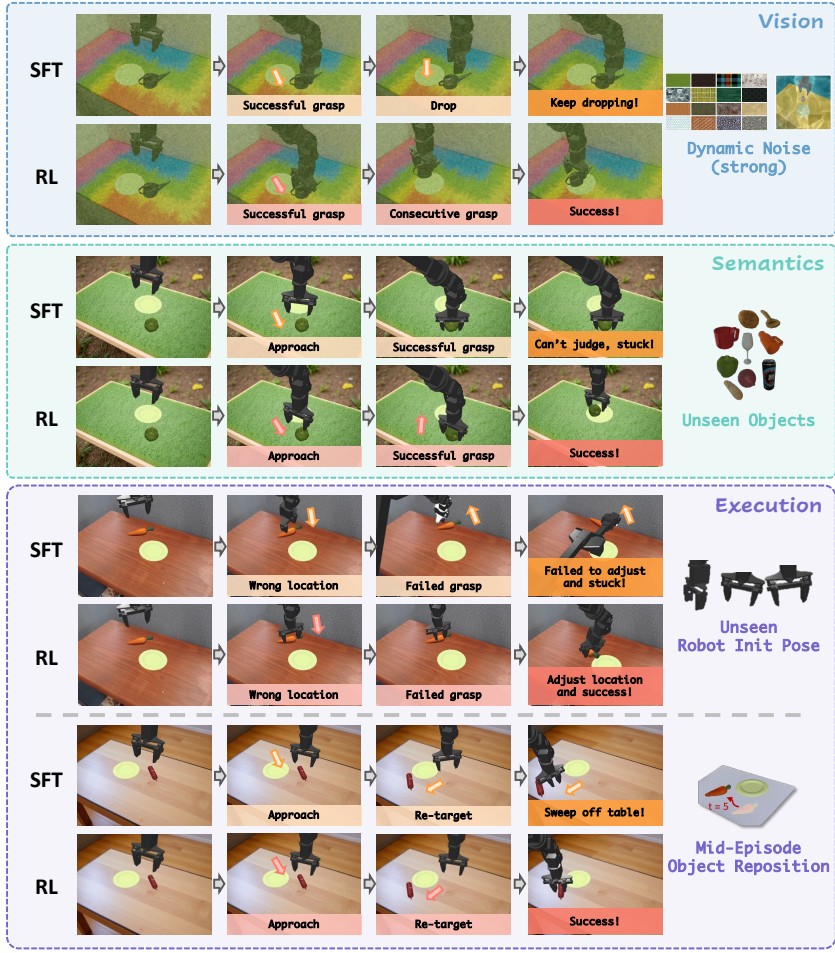

Figure 9: Visualization of some concrete examples.

training (Fig. 8) echo this gap: RL trajectories span a broader workspace and a richer range of end-effector orientations, whereas SFT roll-outs cluster along the motion-planner paths present in the dataset. This wider coverage appears key to RL's superior generalisation on *Execution* tasks.

# 6 Conclusion

This study puts forward a demanding benchmark that probes how RL fine-tuning shapes the out-of-distribution generalization of VLA models. After adapting several RL methods to VLAs, we find that a streamlined PPO variant—featuring a shared actor-critic backbone, a simple warm-up, and only minimal PPO epochs—yields the strongest and most efficient gains. Relative to supervised fine-tuning, this RL approach substantially improves semantic grounding and action execution under distribution shifts, while matching its resilience to visual perturbations. These results underscore the promise of reinforcement learning as a route towards more generalizable embodied agents.

**Limitations.** Despite its contributions, our study has several limitations. First, we rely solely on motion-planner–generated demonstrations for supervised fine-tuning, which may not fully capture the variability present in human-collected data. Second, our evaluation is confined to pick-and-place tasks; scaling to a broader, more complex, multi-task setting remains an important direction for future work. Third, all experiments are conducted in simulation, so integrating RL fine-tuning with sim-to-real transfer to validate VLA generalization on physical robots is a crucial next step. We hope our findings will inform and accelerate these future efforts.

## Acknowledgements

This research was supported by National Natural Science Foundation of China (No.62325405,62406159), Tsinghua University Initiative Scientific Research Program, Tsinghua-Efort Joint Research Center for EAI Computation and Perception, Beijing National Research Center for Information Science, Technology (BNRist), Beijing Innovation Center for Future Chips, State Key laboratory of Space Network and Communications, and Beijing Zhongguancun Academy Project C20250301.

We would like to express our sincere gratitude to Liangzhi Shi, who carried out the deployment on the Franka Panda robotic arm and provided valuable sim-to-real evaluation results. We are also grateful to Hongzhi Zang for her insightful advice on GRPO performance tuning, which significantly improved the quality of this work.

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

# A Implementation Details

## A.1 Implementation of different fine-tuning algorithm

**LoRA finetuning**  Instead of fine-tuning the entire OpenVLA model, we use Low-Rank Adaptation (LoRA) [Hu et al., 2022] to reduce computational costs for all fine-tuning methods in this paper, including both RL and SFT. Specifically, we apply LoRA modules with rank $r = 32$ to all linear layers in the OpenVLA model, while fully fine-tuning the value head.

**PPO update**  The policy is updated by maximizing the following objective [Schulman et al., 2017]

$$L^{\text{CLIP}}(\theta) = \mathbb{E}_t \left[ \min \left( \frac{\pi_\theta(a_t \mid s_t)}{\pi_{\theta_{\text{old}}}(a_t \mid s_t)} \hat{A}_t, \; \text{clip} \left( \frac{\pi_\theta(a_t \mid s_t)}{\pi_{\theta_{\text{old}}}(a_t \mid s_t)}, \; 1 - \epsilon, \; 1 + \epsilon \right) \hat{A}_t \right) \right] \quad (1)$$

where $\pi_\theta(a_t \mid s_t)$ is calculated as the product of the probabilities of each action token, and the advantage $\hat{A}_t$ is given by Generalized Advantage Estimator (GAE) [Schulman et al., 2015], where

$$\hat{A}_t = \sum_{l=0}^{T-t-1} (\gamma\lambda)^l \left[ r_{t+l} + \gamma V(s_{t+l+1}) - V(s_{t+l}) \right] \quad (2)$$

**GRPO implementation**  We follow the outcome supervision RL paradigm in GRPO [Shao et al., 2024], where the advantage is computed as the normalized outcome rewards within each group:

$$\hat{A}_t^i = \frac{r^i - \text{mean}(\mathbf{r})}{\text{std}(\mathbf{r})} \quad (3)$$

$$\mathbf{r} = \{r^1, r^2, \cdots, r^G\} \quad (4)$$

where $r^i$ is the outcome reward for trajectory $i$, and $G$ denotes the number of trajectories in a group.

To stabilize training, we use a total of 32 groups, each containing 8 trajectories. For successful placement episodes, the trajectory is truncated immediately after the first successful step. In contrast, unsuccessful episodes remain untruncated. The outcome reward is given only at the final step of each episode and is defined as the sum of the grasping, holding, and placement rewards.

**DPO implementation**  We employ an improved version of DPO as proposed by [Zhang et al., 2024b], namely Trajectory-wise Preference Optimization (TPO), which fine-tunes the vision-language agent (VLA) by aligning its policy with human preferences over entire trajectories, rather than individual actions. This is achieved by comparing pairs of trajectories (one preferred, one rejected) and optimizing the following loss:

$$\mathcal{L}_{\text{TPO}} = -\mathbb{E}_{(\zeta_w, \zeta_l) \sim \mathcal{D}} \left[ \log \sigma \left( \beta \left( \log \frac{\pi_\theta(\zeta_w)}{\pi_{\text{ref}}(\zeta_w)} - \log \frac{\pi_\theta(\zeta_l)}{\pi_{\text{ref}}(\zeta_l)} \right) \right) \right] \quad (5)$$

where $\zeta_w$ and $\zeta_l$ are the preferred and rejected trajectories, respectively.

In the original TPO paper [Zhang et al., 2024b], the total reward consists of a task success reward, a self-evaluated score, and an LLM-generated dense reward. In our task, only a sparse reward is available, as described in Sec. 5.1. Therefore, we use only this sparse reward for simplicity.

## A.2 Details of motion planner and SFT dataset

We use the "plan_screw" function provided by [Guo et al., 2024]. In detail, we employ a task-space guided iterative inverse kinematics method to generate a locally feasible path that approximates a screw (helical) motion, using the robot's kinematic model (from URDF) and the current joint angles. This path is then post-processed using a Time-Optimal Path Parameterization algorithm (TOPP) [Pham and Pham, 2018], which assigns timing characteristics to the geometric path, resulting in an efficient and executable trajectory (in joint space) that satisfies the robot's kinematic constraints. We perform forward kinematics on the output joint positions and robot current joint positions to obtain the corresponding end-effector poses, and then compute their differences to obtain "delta_ee_pose".

However, data collected by the motion planner contains a fraction of idling actions. When trained with SFT on such data, the fine-tuned OpenVLA model often gets stuck during task execution. To

address this, we employ an action filtering technique: actions with a delta position norm less than $0.01$ and a delta Euler angle norm less than $0.06$ are discarded. Using this filter, approximately one-third of the actions are removed, which significantly alleviates the issue of the model getting stuck. We apply action filtering to all data collected by the motion planner.

### A.3 Details of assets usage

As described in Sec. 5.1, we acquire digital assets from multiple sources. The 3D assets of objects and receptacles are sourced from the ManiSkill simulator [Tao et al., 2024], Objaverse [Deitke et al., 2023], or downloaded from Sketchfab[3]. Additional table appearances are synthesized using Stable Diffusion [Rombach et al., 2022] and ControlNet [Zhang et al., 2023a]; specifically, we use the online service provided by LibLibAI[4]. Distractor textures are downloaded from ambientCG[5]. We comply with the licenses of these assets accordingly.

### A.4 Details of tasks for generalization evaluation

As described in Sec. 5.1, we build multiple tasks to test the generalization of VLA modes. The details of each task is detailed as below:

- *Training*: At the start of each episode, one object from the 16 training objects and one of the 16 training table appearances is selected. The object and the receptacle (yellow plate) are placed on the table, with their positions randomized within a rectangular area on the table. The language instruction is "put $O$ on $R$", where $O$ and $R$ denote the names of the object and receptacle, respectively.

- *Unseen Table*: The table appearance is selected from 5 unseen appearances. Other settings are the same as in the *Training* setting.

- *Dynamic Texture (weak)*: In addition to selecting an object and a table appearance, a texture from the 16 textures is chosen at the start of each episode. This texture is cropped and resized differently at each step in the episode, and is overlaid on top of the object, receptacle, and robot arm in the image, with an image mixing transparency of $0.3$.

- *Dynamic Texture (strong)*: The settings are the same as in the *Dynamic Texture (weak)* setting, except with an image mixing transparency of $0.5$.

- *Dynamic Noise (weak)*: In addition to selecting an object and a table appearance, a texture from the 16 textures is chosen at the start of each episode. This texture is cropped and resized differently at each step in the episode, and is overlaid on whole image, with an image mixing transparency of $0.3$.

- *Dynamic Noise (strong)*: The settings are the same as in the *Dynamic Noise (weak)* setting, except with an image mixing transparency of $0.5$.

- *Unseen Objects*: The object is selected from the 9 unseen objects. Other settings are the same as in the *Training* setting.

- *Unseen Receptacles*: In addition to selecting an object and a table appearance, a receptacle from the 16 unseen receptacles is chosen at the start of each episode, replacing the default training receptacle (yellow plate). Other settings are the same as in the *Training* setting.

- *Unseen Instruction Phrasing*: In addition to selecting an object and a table appearance, a language instruction template from the 16 unseen language templates is chosen at the start of each episode, replacing the default language template ("put $O$ on $R$"). Other settings are the same as in the *Training* setting. The unseen language templates are:
  - Place the $O$ on the $R$
  - set $O$ on $R$
  - move the $O$ to the $R$
  - Take the $O$ and put it on the $R$

---

[3]https://sketchfab.com/
[4]https://liblib.art/
[5]https://ambientcg.com/

- pick up $O$ and set it down on $R$
- please put the $O$ on the $R$
- Put $O$ onto $R$.
- place the $O$ onto the $R$ surface
- Make sure $O$ is on $R$.
- on the $R$, put the $O$
- put the $O$ where the $R$ is
- Move the $O$ from the table to the $R$
- Move $O$ so it's on $R$.
- Can you put $O$ on $R$?
- $O$ on the $R$, please.
- the $O$ should be placed on the $R$.

Note that there are variations in punctuation and capitalization in these instruction templates.

- *Multi-Object (both seen)* At the start of each episode, two different objects from the 16 training objects and one of the 16 training table appearances is selected. These two objects and the receptacle (yellow plate) are placed on the table, with their positions randomized within a rectangular area on the table. The language instruction is "put $O$ on $R$", where $O$ and $R$ denote the names of the fist object and receptacle, respectively.

- *Multi-Object (both unseen)* The two objects are selected from the 9 unseen objects, and they are different from each other. Other settings are the same as in the *Multi-Object (both seen)* setting.

- *Distractive Receptacle*: In addition to selecting an object and a table appearance, a distractive receptacle from the 16 unseen receptacles is chosen at the start of each episode, and placed on the table without being used. Other settings are the same as in the *Training* setting.

- *Multi-Receptacle (both unseen)*: At the start of each episode, an objects from the 16 training objects, two different receptacles from the 16 unseen receptacles, and one of the 16 training table appearances is selected. The object and the two receptacles are placed on the table, with their positions randomized within a rectangular area on the table. The language instruction is "put $O$ on $R$", where $O$ and $R$ denote the names of the object and the first receptacle, respectively.

- *Unseen Position (Object & Receptacle)*: The object and the receptacle (yellow plate) are placed on the table, with their positions randomized within a surrounding square frame, larger than the rectangular area, on the table. Other settings are the same as in the *Training* setting.

- *Unseen Robot Init Pose*: The initial pose of each articulation of the robot is randomized at the start of each episode, rather than being set to a fixed pose as in the *Training* setting. Other settings are the same as in the *Training* setting.

- *Mid-Episode Object Reposition*: The object is teleported to a new random position on the table at the 5-th timestep in an episode. Other settings are the same as in the *Training* setting.

# B   Additional Experiment Results

## B.1   PPO performance on generation temperature

We evaluate the performance of PPO with different generation temperatures during training, as shown in Fig. 10. The results show that a very large temperature hinders the training process, while a moderate or small temperature leads to good performance. Therefore, we use a temperature of 1.0 in PPO.

## B.2   PPO performance on LoRA rank

We evaluate the performance of PPO with different LoRA ranks, as shown in Fig. 11. The results show that a lower LoRA rank leads to slightly more efficient training. In all of our fine-tuning experiments, we use a LoRA rank of 32 to ensure sufficient model capacity. A more detailed study of different LoRA ranks is left for future work.

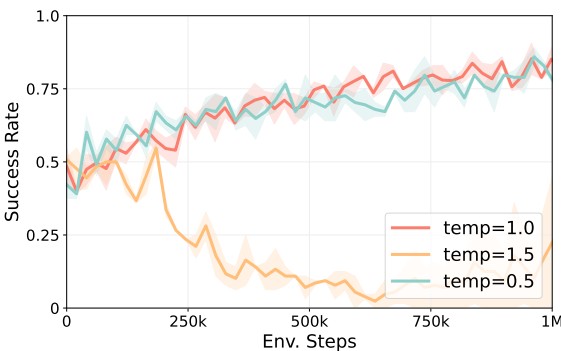

Figure 10: PPO performance on generation temperature during training.

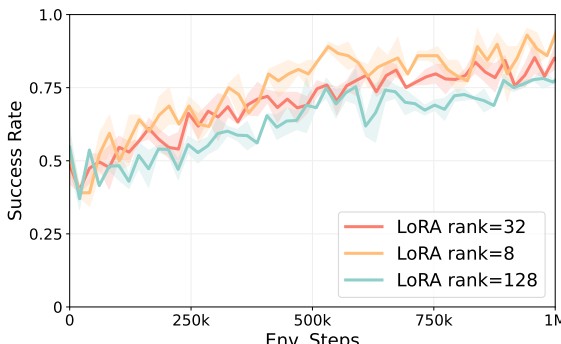

Figure 11: PPO performance on different LoRA rank.

### B.3 Full results of OOD tasks

Detailed results of Fig. 7 are listed in Tab. 1, where grasp accuracy, continuous grasp accuracy and task success rate are reported.

### B.4 Performance comparison on sim-to-real generalization

As discussed, all of our previous experiments are conducted in simulation for scalability and reproducibility when comparing RL and SFT. Nonetheless, real-world performance is a critical factor to validate. We therefore include a preliminary real-world experiment to assess sim-to-real generalizability.

Specifically, we replaced the WidowX arm with a Franka Panda and aligned the camera views. Using the same pipeline as in our main experiments, we retrained VLA models with both RL and SFT under this configuration, and deployed them zero-shot on the real robot. This naturally introduces an OOD setting involving kinematic differences, varied backgrounds, and different real-world objects and receptacles.

In simulation, the training environment contains 16 objects (as described in Sec. 5.1), 7 backgrounds, and a plate as the receptacle. The real-world evaluation involves 6 new objects (pepper, cucumber, banana, mangosteen, kiwi, sponge), a table as the background, and a green bowl as the receptacle. In both simulation and real-world setups, the initial positions of objects and receptacles are randomized. For real-world evaluation, the initial conditions of each trail are aligned across SFT and RL to ensure a fair comparison. The training and evaluation environments are illustrated in Fig. 12.

We report the average performance over 30 real-world trials in Tab. 2. The results show that RL outperforms SFT in both grasp and pick-and-place success rates, demonstrating stronger generalization ability. Qualitatively, we observed that SFT often exhibits excessive movement and overshooting, leading to suboptimal grasp poses and failures. In contrast, RL shows some jitter but successfully iteratively adjusts the end-effector pose, leading to higher grasp success.

Table 1: Detailed results across different tasks, with the best results across SFT and RL highlighted in **bold**. Standard deviations are reported in parentheses.

| | Acc. | | Cont. Acc. | | Suc. | |
|---|---|---|---|---|---|---|
| | SFT | RL | SFT | RL | SFT | RL |
| IND | 0.922 (.013) | **0.974** (.007) | 0.906 (.026) | **0.948** (.007) | 0.781 (.013) | **0.938** (.013) |
| Table | 0.880 (.032) | **0.979** (.007) | 0.828 (.022) | **0.927** (.019) | 0.719 (.051) | **0.844** (.034) |
| Texture-w | 0.896 (.019) | **0.948** (.019) | 0.870 (.039) | **0.906** (.046) | 0.719 (.078) | **0.833** (.048) |
| Texture-s | **0.839** (.019) | 0.802 (.039) | **0.750** (.034) | 0.703 (.066) | 0.557 (.041) | **0.630** (.039) |
| Noise-w | 0.901 (.019) | **0.948** (.027) | 0.854 (.041) | **0.911** (.027) | 0.708 (.032) | **0.854** (.027) |
| Noise-s | 0.781 (.013) | **0.823** (.041) | 0.688 (.034) | **0.750** (.038) | 0.505 (.027) | **0.667** (.048) |
| Obj. | 0.818 (.027) | **0.875** (.000) | 0.724 (.032) | **0.833** (.039) | 0.453 (.044) | **0.714** (.019) |
| Recep. | 0.844 (.013) | **0.911** (.039) | 0.802 (.032) | **0.833** (.019) | 0.615 (.019) | **0.750** (.022) |
| Instruct | 0.885 (.045) | **0.969** (.013) | 0.859 (.044) | **0.948** (.027) | 0.672 (.034) | **0.891** (.034) |
| M-Obj. (IND) | 0.797 (.013) | **0.823** (.027) | 0.771 (.019) | **0.797** (.051) | 0.615 (.052) | **0.750** (.046) |
| M-Obj. (OOD) | 0.656 (.034) | **0.750** (.046) | 0.583 (.019) | **0.688** (.038) | 0.297 (.013) | **0.578** (.034) |
| Disturb Recep. | 0.880 (.032) | **0.948** (.007) | 0.844 (.044) | **0.896** (.027) | 0.672 (.056) | **0.812** (.034) |
| M-Recep. | 0.865 (.019) | **0.885** (.007) | **0.818** (.041) | 0.797 (.034) | 0.458 (.065) | **0.599** (.063) |
| Obj. Pos. | 0.865 (.019) | **0.943** (.007) | 0.802 (.007) | **0.917** (.019) | 0.568 (.070) | **0.807** (.019) |
| Robot Pose | 0.646 (.027) | **0.917** (.063) | 0.568 (.019) | **0.844** (.071) | 0.339 (.052) | **0.797** (.064) |
| Obj. Rep. | 0.573 (.039) | **0.891** (.034) | 0.474 (.060) | **0.807** (.045) | 0.286 (.041) | **0.745** (.048) |

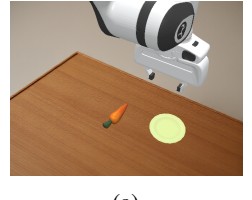 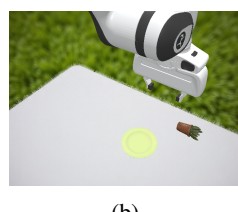 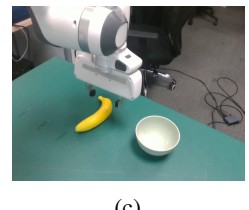 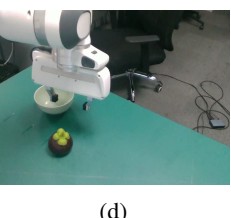

(a)      (b)      (c)      (d)

Figure 12: Visualization of the training and evaluation environments for the preliminary real-world evaluation: (a)-(b) examples of simulation training environments; (c)-(d) examples of real-world evaluation settings.

### B.5 Performance comparison with action chunk

To evaluate the robustness of our conclusions with stronger architectures, we additionally test with OpenVLA-OFT [Kim et al., 2025], which enhance OpenVLA with action chunking.

We adopted an action chunking strategy with a chunk size of 4, enabling the model to predict sequences of actions rather than single steps. To initialize training, we first collected 400 trajectories through motion planning without applying any action filtering, and then trained a warm-up model using a checkpoint from Huggingface[6].

Building on this, we conducted RL fine-tuning in which each observation corresponds to a sequence of 4 actions and is optimized based on cumulative rewards on the following 4 steps. Given the increased dimensionality of the action space, we adjusted the PPO algorithm by introducing action-dimension-wise clipping, reducing the clipping ratio from 0.2 to 0.1, and tuning the hyperparameters with discount factor $\gamma = 0.96$ and GAE $\lambda = 0.85$ to improve training stability.

Results are summarized in Fig. 13 and Tab. 3, showing that RL maintains its advantage over SFT even under this more advanced architecture.

---

[6]Haozhan72/Openvla-oft-SFT-libero10-trajall

Table 2: Preliminary real-world evaluation results. RL fine-tuning in simulation helps sim-to-real transfer compared to SFT.

|  | SFT | RL |
| --- | --- | --- |
| Grasp Success Rate | 0.10 | **0.43** |
| Pick-and-Place Success Rate | 0.00 | **0.27** |

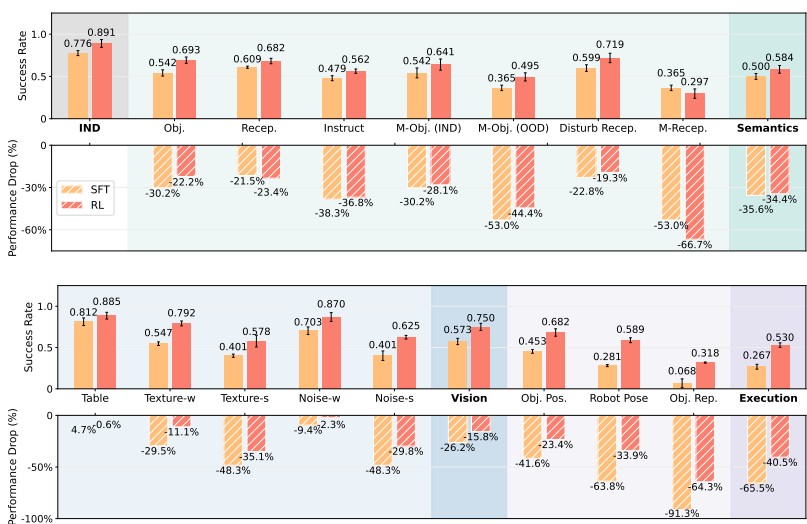

Figure 13: Performance comparison between SFT and RL across different tasks *with action chunking*. Both success rate and relative performance drop are reported.

## B.6 Extended performance comparison on opening articulated objects

To move beyond simple pick-and-place and introduce richer manipulation dexterity, we design an *open articulated objects* task, and replace the WidowX robot arm with a Franka Panda arm to provide a larger working space. The agent must grasp an object's handle and open the door past $20°$. The reward function is shaped as follows: a reward of $0.1$ is given when the handle is first grasped, an additional $0.1$ is granted if the grasp is sustained for five consecutive steps, and a final $1.0$ is provided upon successful door opening.

During training, we introduce randomization across vision, semantics, and execution to improve generalization. For vision, we randomize over 16 background assets (as described in Sec. 5.1). For semantics, we employ 8 articulated objects that vary in size, name, handle type (bar or boxed), and handle position (up, down, left, or right). For execution, we perturb the object frame with random translations up to $\pm 8$ cm and rotations up to $\pm\pi/48$. The training and evaluation environments are illustrated in Fig. 14.

We then evaluate generalization on six out-of-distribution (OOD) splits:

- *Vision:* 5 novel background scenes,

- *Vision:* whole-scene dynamic noise with an image mixing transparency of $0.5$,

- *Semantics:* 4 unseen articulated objects (microwave, dishwasher, mini-fridge, oven),

- *Semantics:* 16 re-phrased language instructions,

- *Execution:* translations up to $12$ cm and rotations up to $\pm\pi/24$,

- *Execution:* random initial offsets on joints 0/1/2/4 of the robot arm.

As shown in Tab. 4, RL continues to outperform SFT on semantic and execution splits, consistent with our main pick-and-place results.

Table 3: Detailed results across different tasks *with action chunking*, with the best results across SFT and RL highlighted in **bold**. Standard deviations are reported in parentheses.

| | Acc. | | Cont. Acc. | | Suc. | |
|---|---|---|---|---|---|---|
| | SFT | RL | SFT | RL | SFT | RL |
| IND | **0.984** (.013) | 0.969 (.022) | **0.953** (.026) | **0.953** (.022) | 0.776 (.029) | **0.891** (.046) |
| Table | **0.964** (.007) | 0.948 (.019) | **0.953** (.013) | 0.948 (.019) | 0.812 (.046) | **0.885** (.041) |
| Texture-w | 0.844 (.046) | **0.885** (.015) | 0.807 (.032) | **0.870** (.007) | 0.547 (.022) | **0.792** (.029) |
| Texture-s | 0.719 (.034) | **0.781** (.056) | 0.661 (.015) | **0.734** (.046) | 0.401 (.019) | **0.578** (.071) |
| Noise-w | 0.932 (.015) | **0.974** (.007) | 0.880 (.019) | **0.958** (.027) | 0.703 (.046) | **0.870** (.053) |
| Noise-s | 0.755 (.039) | **0.839** (.041) | 0.724 (.019) | **0.786** (.045) | 0.401 (.058) | **0.625** (.022) |
| Obj. | 0.807 (.007) | **0.901** (.019) | 0.740 (.007) | **0.875** (.022) | 0.542 (.037) | **0.693** (.037) |
| Recep. | 0.927 (.027) | **0.932** (.045) | 0.885 (.019) | **0.911** (.053) | 0.609 (.013) | **0.682** (.032) |
| Instruct | 0.620 (.007) | **0.641** (.013) | 0.604 (.019) | **0.630** (.019) | 0.479 (.029) | **0.562** (.026) |
| M-Obj. (IND) | 0.661 (.075) | **0.745** (.019) | 0.641 (.080) | **0.714** (.032) | 0.542 (.059) | **0.641** (.066) |
| M-Obj. (OOD) | 0.589 (.048) | **0.661** (.019) | 0.552 (.041) | **0.620** (.019) | 0.365 (.032) | **0.495** (.048) |
| Disturb Recep. | **0.901** (.027) | 0.870 (.029) | **0.891** (.022) | 0.849 (.048) | 0.599 (.039) | **0.719** (.056) |
| M-Recep. | **0.891** (.034) | 0.859 (.038) | **0.849** (.060) | 0.828 (.058) | **0.365** (.032) | 0.297 (.056) |
| Obj. Pos. | 0.807 (.015) | **0.917** (.027) | 0.771 (.015) | **0.896** (.029) | 0.453 (.022) | **0.682** (.045) |
| Robot Pose | 0.557 (.019) | **0.797** (.034) | 0.484 (.034) | **0.734** (.046) | 0.281 (.013) | **0.589** (.029) |
| Obj. Rep. | 0.219 (.038) | **0.510** (.015) | 0.177 (.041) | **0.453** (.013) | 0.068 (.053) | **0.318** (.007) |

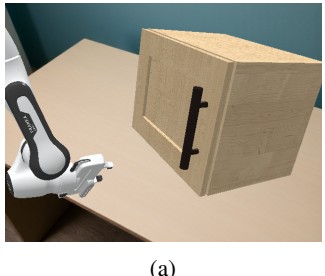 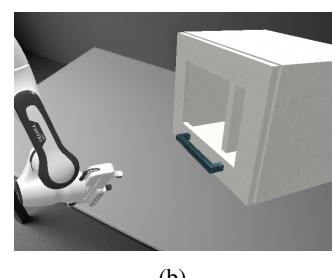 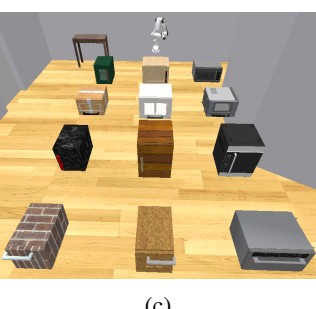

(a)  (b)  (c)

Figure 14: Illustration of the open articulated objects task. (a)-(b) Two representative training scenarios. (c) Set of articulated objects used in training (left two rows) and in OOD testing (the most right row).

## C  Broader impacts

Our work advances the understanding and generalization capabilities of Vision-Language Action (VLA) models in embodied AI, which may accelerate the deployment of more robust and adaptive agents in real-world environments. Improved generalization can enhance the reliability of AI systems in applications such as assistive robotics, autonomous navigation, and household automation, potentially benefiting society by making these technologies safer and more accessible.

However, as VLAs become more capable and generalizable, there is also a risk of unintended consequences, such as misuse in surveillance, privacy violations, or deployment in safety-critical scenarios without adequate oversight. We encourage responsible use of our findings and recommend that future deployments of VLA-based systems consider safety, ethical, and privacy implications.

Table 4: Performance comparison on the open articulated objects task, with results averaged over 3 random seeds. Standard deviations are reported in parentheses.

| | SFT | | RL | |
|---|---|---|---|---|
| | Success Rate | Degradation Ratio | Success Rate | Degradation Ratio |
| IND | **0.745** (.015) | – | 0.724 (.045) | – |
| New Backgrounds | **0.766** (.071) | **+0.028** | 0.734 (.058) | +0.014 |
| Dynamic Noise | **0.526** (.015) | **-0.294** | 0.448 (.037) | -0.381 |
| **Vision Avg.** | **0.646** (.043) | **-0.130** | 0.591 (.048) | -0.184 |
| Articulated Obj. | 0.042 (.015) | -0.944 | **0.151** (.019) | **-0.791** |
| Re-phrased Inst. | **0.703** (.077) | **-0.056** | 0.620 (.041) | -0.144 |
| **Semantic Avg.** | 0.373 (.046) | -0.500 | **0.386** (.030) | **-0.468** |
| Obj. Pose | 0.448 (.032) | -0.399 | **0.484** (.089) | **-0.331** |
| EE Pose | 0.151 (.027) | -0.797 | **0.312** (.013) | **-0.568** |
| **Execution Avg.** | 0.300 (.030) | -0.600 | **0.398** (.051) | **-0.450** |

