# OpenReview forum: "What Can RL Bring to VLA Generalization? An Empirical Study"
_NeurIPS.cc/2025/Conference — NeurIPS 2025 poster_

### Official Review · Reviewer_vvPb · 2025-06-10

**Clarity:** 3
**Significance:** 2
**Originality:** 3
**Rating:** 5
**Confidence:** 4

**Summary:**

The authors propose a novel benchmark to text the generalization capabilities of Vision-Language Action (VLA) models. They are able to text the generalization capabilities of the VLA models along the vision, semantics and execution axis. The authors then test the performance of SFT and a few RL algorithms on different OOD dimensions. They found that PPO generalizes better than SFT and some RL baselines for semantic understanding and execution robustness, while it matches the baselines for visual tasks. The authors also provide an insightful analysis of the exploration performed by RL in Figure 8.

**Questions:**

* For PPO epoch >1, do you recompute the advantages using the updated value function?
* Can you compare your study to the ones reported in Figure 4 of RT2?

**Ethical Concerns:**

["NO or VERY MINOR ethics concerns only"]

**Final Justification:**

This paper introduced an interesting benchmark to evaluate Vision-Language Action models on out of distribution tasks. While doing a literature review, I was surprised that RL has not been widely explored by previous work. Therefore, I think their contribution is worth accepting to the conference.

**Limitations:**

* The authors mostly compared PPO against algorithms designed for single step LLM training. It is not surprising that they do not perform very well on multi-step robotic tasks. A more interesting comparison would be to compare against RL algorithms such as A3C, TD3, VMPO, etc.

**Quality:**

3

**Strengths And Weaknesses:**

# Strengths

## Quality
* Well designed benchmark to test OOD performance along controllable axis: vision, semantics, and execution. The field needs more carefully designed empirical work like this.
* Multiple seeds (2 and 3) are used. While more seeds could be used, it is a big improvement over a single seed.

## Novelty
* Surprisingly most VLA are not trained via RL e.g. RT2, \pi_0, Octo, etc. By showing the advantages of PPO over the SFT baseline, this work motivates further investigation for training VLA with RL methods.

## Impact
* The authors provide an efficient training recipe (42h on a A100 gpu) and several ablation (shared backbone, val warm up and number of ppo epochs) for the PPO design choices. This simple and efficient recipe is impactful will allow other academic lab to build on this work.

## Visualization
* The visualization of the training trajectories in Figure 8 is interesting.

# Weaknesses

## Comprehensiveness
* The authors test a single model (OpenVLA) and a single task (pick and place).

## Significance
* The main takeaway is not surprising: RL algorithms designed for multi-step tasks (PPO) perform better on multi-step tasks than algorithms designed for single-step tasks (GRPO and DPO). I think the paper would be more interesting if the RL algorithms used for comparison were carefully chosen, e.g. soft actor critic, etc.


## Clarity
* In Figure 1, I am not sure that the emojis are very useful nor what they imply. It seems like the OOD RL performance drops similarly across vision, semantics and execution, but 3 different emojis are used.

---

> ### Author Rebuttal · Authors · 2025-07-31
>
> We sincerely appreciate the reviewer’s supportive comments on our strengths across the quality, novelty, impact, and visualization. Below, we address your remaining concerns clearly and thoroughly.
>
> > ### Q1: Extend to more VLA models and more complex tasks
>
> Thank you for raising this important concern. While our original submission focused on a single model (OpenVLA) and task (pick-and-place) to establish a controlled foundation for comparison, we have since expanded our study in two complementary directions to reinforce the generality of our findings.
>
> **First, to evaluate broader VLA architectures**, we conducted additional experiments on **OpenVLA-OFT** [3], a stronger variant of OpenVLA that incorporates **action chunking**. This model poses increased challenges for RL training due to its higher-dimensional, temporally extended action space. We adapted our RL pipeline accordingly—modifying reward computation, clip ratio, and advantage estimation—and observed consistent improvements from RL fine-tuning over SFT. These results indicate that the benefits of RL are not confined to the original model and extend to more complex action representations.
>
> |Tasks|SFT Success Rate (3 seeds)|SFT Decline Ratio|RL Success Rate (3 seeds)|RL Decline Ratio|
> |-|-|-|-|-|
> |In Distribution|0.536||0.891||
> ||||||
> |Table|0.568|**0.058**|0.885|-0.006|
> |Texture-w|0.438|-0.184|0.792|**-0.111**|
> |Texture-s|0.328|-0.388|0.578|**-0.351**|
> |Noise-w|0.464|-0.136|0.870|**-0.023**|
> |Noise-s|0.349|-0.35|0.625|**-0.298**|
> |**Vision Avg.**|0.429|-0.2|0.75|**-0.158**|
> ||||||
> |Obj.|0.438|**-0.184**|0.693|-0.222|
> |Recep.|0.349|-0.35|0.682|**-0.234**|
> |Instruct|0.318|-0.408|0.562|**-0.368**|
> |M-Obj. (IND)|0.438|**-0.184**|0.641|-0.281|
> |M-Obj. (OOD)|0.234|-0.563|0.495|**-0.444**|
> |Disturb Recep.|0.432|-0.194|0.719|**-0.193**|
> |M-Recep.|0.182|**-0.66**|0.297|-0.667|
> |**Semantic Avg.**|0.342|-0.363|0.584|**-0.344**|
> ||||||
> |Obj. Pos.|0.266|-0.505|0.682|**-0.234**|
> |Robot Pose|0.188|-0.65|0.589|**-0.339**|
> |Obj. Rep.|0.057|-0.893|0.318|**-0.643**|
> |**Execution Avg.**|0.17|-0.683|0.53|**-0.405**|
>
> **Second, to evaluate more complex tasks**, we introduced a more challenging **open-cabinet manipulation** benchmark. This task requires more dexterous manipulation with a articulated object, including handle grasping and door opening under significant semantic and geometric variability. Across six out-of-distribution splits—including novel objects, diverse cabinet configurations, and perturbed initial states—RL-fine-tuned policies again consistently outperformed SFT, especially on **execution** and **semantic generalization**. This echoes the trend observed in the pick-and-place setting and provides further evidence that RL enhances robustness across manipulation tasks.
>
> |splits|SFT Success Rate (3 seeds)|SFT Decline Ratio|RL Success Rate (3 seeds)|RL Decline Ratio|
> |-|-|-|-|-|
> |In Distribution|0.745||0.724|-|
> ||||||
> |New Backgrounds|0.766|**0.028**|0.734|0.014|
> |Dynamic Noise|0.526|**-0.294**|0.448|-0.381|
> |**Vision Avg.**|0.646|**-0.13**|0.591|-0.1835|
> ||||||
> |Articulated Obj.|0.042|-0.944|0.151|**-0.791**|
> |Re‑phrased Inst.|0.703|**-0.056**|0.620|-0.144|
> |**Semantic Avg.**|0.373|-0.5|0.386|**-0.4675**|
> ||||||
> |Obj. Pose|0.448|-0.399|0.484|**-0.331**|
> |EE Pose|0.151|-0.797|0.312|**-0.568**|
> |**Execution Avg.**|0.3|-0.6|0.398|**-0.4495**|
>
> While each extension targets a different axis (model or task complexity), both strengthen our core conclusion: **RL fine-tuning offers more robust generalization than SFT, particularly in execution and semantic variation**, and this advantage holds across models and tasks of increasing complexity.
>
> > ### Q2: Extended to more RL algorithms like SAC
>
> We sincerely thank the reviewer for this thoughtful suggestion. We agree that evaluating our approach with a standard off-policy algorithm such as SAC would be a valuable addition to the study and could further enrich our comparative analysis.
>
> That said, we would like to emphasize that our current comparisons among PPO, GRPO, and DPO still offer meaningful contributions. GRPO and DPO represent recent advances specifically designed to fine-tune policies from demonstrations and preferences—both widely discussed in the context of language and vision-language models. Our finding that PPO, an on-policy RL algorithm, outperforms GRPO and DPO on embodied multi-step tasks is not merely expected—it provides a concrete empirical validation that these methods may fall short when applied to sequential VLA control tasks. To our knowledge, this is the first systematic comparison of these algorithms on such models, and the insights gained are directly actionable for researchers adapting RL methods to VLA domains.
>
> Regarding the inclusion of SAC, we acknowledge its relevance and have actively considered it. However, applying SAC to our setting is non-trivial, as OpenVLA operates in a discrete, high-dimensional action space: 7 dimensions with 256 bins each. This results in an effective action space of 256⁷ combinations, making traditional continuous-action SAC inapplicable without substantial adaptation. One workaround involves Gumbel-Softmax-based relaxation to approximate gradients for discrete actions, but this results in a 1792-dimensional action input to the Q-network, posing severe challenges for learning a stable and expressive critic.
>
> Despite these challenges, we are working diligently to explore SAC-based experiments within the remaining rebuttal/discussion period and will incorporate complete results in a revised version of the manuscript. We appreciate the reviewer’s suggestion and view it as a valuable direction for future iterations of this work.
>
> > ### Q3: Implication of emojis
>
> Thank you for pointing this out, and we apologize for any confusion caused by the use of emojis in Figure 1. Our intent was not to suggest the absolute performance drop of RL in each OOD setting, but rather to highlight the relative performance of RL compared to SFT across the three axes. Specifically, the emojis were chosen to reflect: *in out-of-distribution tests, RL yields substantial gains in Execution, moderate improvements in Semantics, and performance on par with SFT for Vision.*
>
> We used distinct emojis to visually summarize this pattern of relative advantage, not to indicate differences in absolute performance degradation. We appreciate the reviewer’s feedback and will revise the figure caption and visual elements in the revised version to make this intent clearer.
>
> > ### Q4: Advantage computation in PPO
>
> We **did NOT** recompute advantages after the first pass. In PPO, a rollout is collected with the current policy, GAE is computed once with the value-function of that policy, and the resulting advantages are kept fixed while the same data are reused for several mini-batch updates across later epochs. This is also how many well-known RL codebases, like Stable-Baselines3, MAPPO and "37 Implementation Details of PPO" blog, handle PPO updates, so our implementation follows common practice.
>
> > ### Q5: Comparison between ours and RT2's Fig4 experiments
>
> Although both our study and the RT-2 experiments in Fig. 4 evaluate policies under several OOD settings, the research questions diverge:
>
> - **Our focus** is to ask *"What unique benefits can RL bring to VLA generalization compared to SFT?"*. Hence, we start from the **same pretrained VLA backbone** and compare an RL-fine-tuned policy with its SFT counterpart.
> - **RT-2’s focus** is to show that *web-scale pre-training endows VLA models with broadly transferable visual and semantic concepts*. Their evaluation therefore contrasts **different model architectures, data sources, and training pipelines**.
>
> Because of these distinct aims and setups, the two studies are complementary rather than directly comparable. Moreover, our benchmark spans three orthogonal generalization dimensions, offering a more systematic evaluation suite than RT-2's three unseen tests.

---

> > ### Comment · Reviewer_vvPb · 2025-08-05
> > **Response to rebuttal**
> >
> > Thank you for your comment and expanding the tasks. This is a great contribution to the community. I will maintain my score.

---

### Official Review · Reviewer_84hZ · 2025-06-30

**Clarity:** 3
**Significance:** 3
**Originality:** 3
**Rating:** 4
**Confidence:** 3

**Summary:**

The paper can be split into two parts. The first part shows that for RL fine-tuning VLAs, PPO is more effective than LLM-derived methods like DPO, GRPO, and explores design choices of PPO such as critic design, warm-up, and UTD tuning. The second shows that RL tuning (with PPO) has superior generalization capabilities compared to supervised fine-tuning (SFT). For this, the authors design a novel benchmark (based on Maniskill) to systematically evaluate generalizability on novel vision, novel semantics (e.g., unseen objects), and novel execution (e.g., random initial state).

**Questions:**

1. In Figure 6a and 6b, the performance does not plateau after 16k but visibly drops on success rate. Why do you think this happened?

**Ethical Concerns:**

["NO or VERY MINOR ethics concerns only"]

**Final Justification:**

I think the paper has enough contribution to the community to be accepted as a paper (dataset and pipeline for evaluating generalization of VLAs). However, the overall structure and quality of the paper holds me from giving a higher score. Although I came to agree that successfully training VLA with RL can be a contribution, I still believe it is not a significant one.

**Limitations:**

Yes

**Quality:**

3

**Strengths And Weaknesses:**

Strengths
- Straightforward and concrete experimental setup, and strong results.
- Extensive investigation on VLA's generalization in pick-and-place with diverse out-of-domain scenarios, accompanied with interesting analyses on the results (figure 7,8,9).

Weaknesses
- The paper spends a great amount of writing space to PPO itself, which seems redundant to the main topic (generalization). I think the paper would have been more coherent if the design choices were investigated on their effect to generalization capabilities (although I'm not sure about the connection between generalization and the components in Section 4.2).
- The dynamic texture and dynamic noise (applying flat texture noise to the robot/object or to the entire view) design seems a bit unrealistic; I doubt that those visual noises will actually occur in practice.
- As mentioned in the limitation, the quality of the SFT dataset is far from satisfactory (figure 8a), which raises questions on the results (especially that of figure6a,b).

---

> ### Author Rebuttal · Authors · 2025-07-31
>
> We sincerely thank the reviewers for their valuable feedback and positive evaluation of our research. Below, we provide detailed responses to each of your comments.
>
> > ### Q1: Meaning of our PPO recipe and experiments
>
> Thank you for your attention to the desciptions devoted to PPO and Section 4.2. We understand the concern and appreciate the opportunity to clarify its role in the paper.
>
> When we conducted this study, applying PPO to finetune a VLA model remained a non-trivial and open problem. Although there have been some emerging concurrent work [1,2] on PPO+VLA since our submission, our work is one of initial studies in this direction. There was no established or off-the-shelf RL method that could directly be used to study generalization in our study. As such, developing an effective PPO training recipe was necessary to enable a systematic and fair comparison of generalization between RL and SFT. Without this foundation, it would be difficult to make credible claims about RL’s potential contributions to generalization.
>
> Moreover, the training decisions described in Section 4.2 are not standard or trivial; through empirical tuning, we identified several key design choices that significantly impact performance and training efficiency. These insights are valuable in their own right and can guide future work on applying RL to VLA models.
>
> Besides, the effects of these design choices are not isolated to in-distribution performance but show consistent relative performance for out-of-distribution tests. Thus, we base the rest of our comparative experiments on this PPO recipe.
>
> We will make these motivations and the connection to generalization more explicit in the revised version to improve clarity and coherence.
>
> [1] Interactive Post-Training for Vision-Language-Action Models. arXiv:2505.17016
>
> [2] VLA-RL: Towards Masterful and General Robotic Manipulation with Scalable Reinforcement Learning. arXiv:2505.18719
>
> > ### Q2: Is it realistic and practical to apply visual noise as done in our experiments?
>
> Thank you for raising this concern. Stress-testing vision with deliberately extreme textures or lighting is already standard practice for gauging a manipulation policy’s perceptual invariance.
> - **Purpose-built stress tests are widely accepted**. The DMControl generalization benchmark [3] suite overlays randomly sampled textures and colours on every frame to create out-of-distribution (OOD) views; many works like [4] evaluate policy performance under these texture-shifted test sets, treating them as a proxy for unexpected backgrounds and camera disturbances.
> - **Real-robot studies adopt the similar idea**. Prior work on manipulation like [5] evaluates the same policy after changing table, background colours, camera viewpoints, and dynamic Disco lighting conditions, demonstrating that such appearance swaps are considered a practical robustness check.
> - **Community benchmarks adopt similar disturbed evaluation**. The Colosseum benchmark [6] includes “Texture Swap” and “Lighting Perturbation” splits in which object and table surfaces, as well as global illumination hues, are randomised at test time; leading methods are ranked by how gracefully they handle these overlays.
>
> Our Dynamic-Texture and Dynamic-Noise splits follow the same philosophy: although the specific patterns may be unlikely to appear in daily operation, they create a controlled yet harsh visual perturbation that reveals whether a policy has learned stable, object-centric features rather than over-fitting to background colour or illumination. Therefore, the evaluation protocol is both practical and comparable to widely adopted benchmarks.
>
> [3] Generalization in reinforcement learning by soft data augmentation. ICRA 2021
>
> [4] Generalizable Visual Reinforcement Learning with Segment Anything Model. arXiv:2312.17116
>
> [5] Learning to Manipulate Anywhere: A Visual-Generalizable Framework for Reinforcement Learning. CoRL 2024
>
> [6] The Colosseum: A Benchmark for Evaluating Generalization for Robotic Manipulation. RSS 2024
>
>
> > ### Q3: The quality of the SFT dataset
>
> Thank you for your insightful question. We acknowledge the quality limitation of our SFT dataset as mentioned in the paper. To address this concern, we have taken steps to improve data diversity by tuning our motion planner to explore a larger state space. Additionally, as noted in Section 4.2 (lines 213-214), we experimented with combining motion-planned data with demonstrations collected using a pretrained policy (e.g., Octo). However, our experiments revealed that the performance difference between motion-planned and Octo-collected data was minimal for SFT. Therefore, considering scalability advantages, we adopted the motion planner approach to collect up to 64k trajectories.
>
> Regarding the performance drop after 16k trajectories in Figures 6a and 6b, we hypothesize that with larger datasets, it becomes increasingly challenging for the policy to effectively cover all training samples, potentially leading to the observed performance degradation. It's worth noting that while the success rate shows a decline, the grasp accuracy and continuous grasp accuracy metrics appear to plateau, suggesting that the policy maintains certain capabilities despite the overall performance drop. This phenomenon warrants further investigation, which we plan to investigate further in future work.

---

### Official Review · Reviewer_ePmx · 2025-07-01

**Clarity:** 3
**Significance:** 2
**Originality:** 3
**Rating:** 3
**Confidence:** 4

**Summary:**

This paper introduces a new benchmark to test the generalization capabilities of VLAs on a pick-and-place task, focusing on three factors: 1) visual appearance generalization, 2) semantic-level generalization, and 3) execution level generalization. It picks OpenVLA as the backbone model, and performed a thorough empirical study over several RL algorithms, including PPO, GRPO, and DPO, compared with SFT. The paper draws the conclusions that 1) RL works better in semantics and execution, but performs equally well in vision; 2) PPO works the best. It also gives insights about the design choices of PPOs for better performance in RL.

**Questions:**

1. Is it possible to verify the conclusions on a real-robot? Even a simple task would be helpful.

2. Have the authors tried a more advanced policy, e.g., pi0?

3. Have the authors tried a standard off-policy RL algorithm, e.g. SAC / MPO, instead of the methods from RLHF literature?

**Ethical Concerns:**

["NO or VERY MINOR ethics concerns only"]

**Final Justification:**

I think the paper do have some merit, as also agreed by other reviewers. However, I'm still concerned about its way of running experiments. On-policy RL algorithms have natural limitations when applied to robotics, and I think from the very beginning, choosing this family of algorithms is a less reasonable decision choice. However, I also feel this work has certain merits and I'm not opposed to an acceptance.

**Limitations:**

yes

**Quality:**

3

**Strengths And Weaknesses:**

Strength

This paper studies an interesting and important question: what can RL bring to VLA generalization, which from my point of view, is an important topic the community will have to work on in the future. It has also designed a few meaningful tasks and performed a thorough study over the algorithms.

Weaknesses

There are a few weaknesses of the work and I’m very concerned about whether the results and insights can be transferred to real-world settings.

1. The biggest issue is that the findings are drawn from an on-policy RL algorithm, PPO. However, it is extremely challenging to deploy any on-policy RL algorithms on a real robot due to the poor sample efficiency. In Fig. 6, all plots need 2m steps, which is merely impossible for a real robot. In addition, the only applicable algorithm, DPO, doesn’t seem to give any performance gain.
2. The other issue is the choice of the base model, OpenVLA. OpenVLA predicts a single-step action per time-step. In contrast, latest VLA policies often predicts an “action chunk” which consists of multi-step predictions. Given an action-chunk policy, it is possible that modifications to the RL algorithms are required to make it work. Besides, these VLA models often rely on diffusion policies / flow matching policies, which might require extra efforts during RL.
3. The benchmark is overly simplified as a pick-place task. The conclusions can be hard to transfer to tasks that require more dexterous skills, e.g., folding clothes.

---

> ### Author Rebuttal · Authors · 2025-07-31
>
> Thank you for your insightful comments and for highlighting these important considerations. We appreciate the opportunity to further clarify the goals and contributions of our paper.
>
> We acknowledge that real-world RL for VLA and advanced versions with flow matching are still in early development stages. Our focus, therefore, is not on achieving SOTA performance, but on providing a thorough comparative analysis of RL versus SFT that offers insights applicable across different methods and models.
>
> To this end, we **strategically chose a simulation-based evaluation** for two compelling reasons:
>
> - **Rigorous benchmarking**: Large-scale controlled simulations are essential for comprehensive ablation studies, reproducibility, and broad scenario coverage—critical requirements for establishing a benchmark with lasting research impact. Real-world trials alone are hard to provide this level of systematic analysis.
> - **Intrinsic merit**: Simulation-based RL training has its own value, especially given ongoing debates regarding whether simulation-based training can enhance VLA's abilities. Concurrent works [1,2] have also demonstrated the potential advantages of on-policy RL trained in simulated environments. To address further concerns, **we have included preliminary real-world results validating sim-to-real transfer** during this rebuttal period (detailed in our Q1 response).
>
> To support the generalizability of our findings, we have conducted extensive additional experiments, including tests with alternative VLA architectures, more complex tasks, and off-policy RL algorithms. These results consistently validate our core findings about the distinct effects of RL in comparison to SFT.
>
> [1] Interactive Post-Training for Vision-Language-Action Models. arXiv:2505.17016
>
> [2] VLA-RL: Towards Masterful and General Robotic Manipulation with Scalable Reinforcement Learning. arXiv:2505.18719
>
> Below, we will address your specific questions.
>
> > ### Q1: Additional experiments to verify the conclusions on a real-robot
>
> As discussed, our experiments are intentionally conducted in simulation for scalability and reproducibility when comparing RL and SFT. Nonetheless, we fully agree that real-world performance is a critical aspect to validate and have therefore included a preliminary real-world experiment to assess sim-to-real generalizability—adding an additional generalization dimension to our evaluation.
>
> According to our real-world setups, we adapted the simulation by replacing the WidowX arm with a Franka Panda and aligning the camera views. Using the same pipeline as in our main experiments, we re-trained VLA models with both RL and SFT under this new configuration and deployed them zero-shot on the real robot. This setting introduces a challenging OOD setting involving kinematic differences between the simulated and real arms, varied backgrounds, and different objects (including pepper, cucumber, banana, mangosteen, kiwi, and sponge) and receptacles (in specific, a green bowl). We report the average performance over 30 real-world trials in this table:
>
> ||Success Rate for Grasping|Success Rate for full Pick-and-Place|
> |-|-|-|
> |RL|**0.43**|**0.27**|
> |SFT|0.1|0.0|
>
> As shown in the table, RL fine-tuning in simulation can also help the sim-to-real generalization compared to SFT. We observed that when approaching the object, SFT often exhibits excessive movement and tends to overshoot, resulting in a suboptimal grasping pose and subsequent grasp failures. In contrast, RL, despite showing some jitter in its actions, is able to iteratively adjust its grasping pose around the object. This leads to a higher grasp success rate.
>
> > ### Q2: Additional experiments with a standard off-policy RL algorithm (e.g. SAC)
>
> We sincerely thank the reviewer for this suggestion. We agree that evaluating our approach with a standard off-policy algorithm like SAC would further strengthen our work.
>
> However, applying SAC to OpenVLA is non-trivial, especially within the limited rebuttal period. Our VLA model features a discrete, high-dimensional action space (7 dimensions, each with 256 bins), requiring techniques like the Gumbel-Softmax trick for differentiability. This results in a very high-dimensional (256 × 7) action input for the Q-network, making stable learning particularly challenging.
>
> Despite these technical hurdles, we are trying our best to conduct these experiments and hope to provide these additional results before the discussion period concludes. **We are fully committed to incorporating the complete results and a detailed analysis of the SAC experiments into a revised version of our paper.** We appreciate the reviewer's valuable feedback to further complement our study.
>
> > ### Q3: Additional experiments with a more advanced VLA model
>
> Thank you for raising this insightful question regarding the choice of our base model. We agree that advanced VLA architectures, such as Pi-0, which incorporate diffusion or flow-matching policies, indeed pose unique challenges for RL fine-tuning. To date, end-to-end RL methods for models like Pi-0 have not been convincingly demonstrated in the literature, and developing such methods would extend beyond the scope of our systematic comparative study.
>
> Nevertheless, we acknowledge the importance of evaluating our conclusions with a more advanced VLA architecture. To address your concern, we have introduced an additional model, OpenVLA-OFT [3], which enhances the original OpenVLA model by integrating action chunking. To adapt our RL pipeline for OpenVLA-OFT, we made the following modifications:
> 1. **Action Chunking:** We utilized an action chunk size of 4.
> 2. **Warm-up Model Training:**
>    - Collected 400 trajectories via motion planning without action filtering.
>    - Trained a warm-up model from `Haozhan72/Openvla-oft-SFT-libero10-trajall` (a checkpoint from Huggingface).
> 3. **RL Fine-tuning Adjustments:**
>    - Each observation predicts a sequence of 4 actions, with cumulative rewards.
>    - To accommodate the higher dimensional action space, we modified the PPO clipping mechanism to perform action-dimension-wise clipping and reduced the clip ratio from 0.2 to 0.1, and set gamma to 0.96 and lambda to 0.85 for stability.
>
> Results show that RL maintains its advantage over SFT even with this more advanced architecture:
>
> |Tasks|SFT Success Rate (3 seeds)|SFT Decline Ratio|RL Success Rate (3 seeds)|RL Decline Ratio|
> |-|-|-|-|-|
> |In Distribution|0.536||0.891||
> ||||||
> |Table|0.568|**0.058**|0.885|-0.006|
> |Texture-w|0.438|-0.184|0.792|**-0.111**|
> |Texture-s|0.328|-0.388|0.578|**-0.351**|
> |Noise-w|0.464|-0.136|0.870|**-0.023**|
> |Noise-s|0.349|-0.35|0.625|**-0.298**|
> |**Vision Avg.**|0.429|-0.2|0.75|**-0.158**|
> ||||||
> |Obj.|0.438|**-0.184**|0.693|-0.222|
> |Recep.|0.349|-0.35|0.682|**-0.234**|
> |Instruct|0.318|-0.408|0.562|**-0.368**|
> |M-Obj. (IND)|0.438|**-0.184**|0.641|-0.281|
> |M-Obj. (OOD)|0.234|-0.563|0.495|**-0.444**|
> |Disturb Recep.|0.432|-0.194|0.719|**-0.193**|
> |M-Recep.|0.182|**-0.66**|0.297|-0.667|
> |**Semantic Avg.**|0.342|-0.363|0.584|**-0.344**|
> ||||||
> |Obj. Pos.|0.266|-0.505|0.682|**-0.234**|
> |Robot Pose|0.188|-0.65|0.589|**-0.339**|
> |Obj. Rep.|0.057|-0.893|0.318|**-0.643**|
> |**Execution Avg.**|0.17|-0.683|0.53|**-0.405**|
>
> [3] Fine-Tuning Vision-Language-Action Models: Optimizing Speed and Success. RSS 2025.
>
> > ### Q4: Additional experiments on more complex tasks
>
> Thank you for highlighting the need for a task that demands richer dexterity than pick‑and‑place. To address this, we extended the benchmark with an **open‑cabinet task** that requires grasping a handle and opening the door past 20 °. The reward function assigns 0.1 when the handle is first grasped, 0.1 when the grasp is sustained for five consecutive steps, and 1.0 upon successful door opening. During training, we randomized:
> - Vision: 16 backgrounds from the original assets.
> - Semantics: 8 cabinets of varying sizes, names, handle types (bar, boxed) and handle positions (up, down, left, right).
> - Execution: Cabinet frame translations (±8 cm) and rotations (±π/48).
>
> We then test on **six OOD splits**: (i) 5 new backgrounds, (ii) whole‑scene dynamic noise, (iii) 4 novel articulated objects (e.g., microwave, mini‑fridge), (iv) 20 re‑phrased language instructions, (v) cabinet translations up to 12 cm with rotations up to ±$\pi$/24, and (vi) random initial offsets on joints 0/1/2/4 of the end‑effector.
>
> Among all splits, RL fine-tuning outperforms SFT on execution and semantic OOD tests, and performs comparably on vision, consistent with our findings on the pick-and-place task.
>
>
> |splits|SFT Success Rate (3 seeds)|SFT Decline Ratio|RL Success Rate (3 seeds)|RL Decline Ratio|
> |-|-|-|-|-|
> |In Distribution|0.745||0.724|-|
> ||||||
> |New Backgrounds|0.766|**0.028**|0.734|0.014|
> |Dynamic Noise|0.526|**-0.294**|0.448|-0.381|
> |**Vision Avg.**|0.646|**-0.13**|0.591|-0.1835|
> ||||||
> |Articulated Obj.|0.042|-0.944|0.151|**-0.791**|
> |Re‑phrased Inst.|0.703|**-0.056**|0.620|-0.144|
> |**Semantic Avg.**|0.373|-0.5|0.386|**-0.4675**|
> ||||||
> |Obj. Pose|0.448|-0.399|0.484|**-0.331**|
> |EE Pose|0.151|-0.797|0.312|**-0.568**|
> |**Execution Avg.**|0.3|-0.6|0.398|**-0.4495**|
>
> ---
> ---
> We deeply appreciate your thorough review and constructive feedback, which has enabled us to strengthen our work substantially. We are truly encouraged by your recognition of our study as “thorough”, as our primary goal was to provide the community with **a systematic and reproducible empirical comparison of RL and SFT**, rather than to propose a new state-of-the-art algorithm.
>
> In response to your concerns, we have conducted a wide range of new experiments within the limited rebuttal period. We believe that these additional results further enhance the integrity of our study. **We kindly ask you to reconsider your evaluation in light of our clarifications and the new experiments.**

---

> > ### Comment · Reviewer_ePmx · 2025-08-04
> >
> > I appreciate the authors' detailed response. Part of my concerns regarding action chunking has been addressed. However, I do think playing with on-policy RL is less meaningful to robotics, given its current limitations. I will slightly increase my score to 3.

---

> > > ### Author Response · Authors · 2025-08-04
> > >
> > > Thank you for raising your score from **2 → 3**. However, we remain concerned that your overall verdict still trends toward rejection because you regard “playing with on-policy RL [as] less meaningful to robotics.” **We respectfully disagree and would like to address this point head-on**.
> > >
> > > On-policy algorithms such as PPO underlie several landmark results: OpenAI’s Rubik’s-cube hand [1], agile quadruped locomotion learned in minutes [2], humanoid motion via DeepMimic [3], and the latest VLA agents fine-tuned with on-policy RL [4, 5]. **These successes show that on-policy RL is already central to robotics practice.**
> > >
> > > We chose PPO because (i) it is the state-of-the-art baseline in our large-scale simulation regime and (ii) modern sim-to-real techniques already let us deploy PPO-trained policies, as evidenced by existing works listed above and our initial hardware results provided in the previous rebuttal; real-world RL can then refine these policies further. **Our study therefore complements rather than competes with off-policy or real-world RL work.**
> > >
> > > In this light, we kindly ask you to reassess your evaluation based on the paper’s core contribution: **a controlled study that isolates RL’s unique gains over SFT for VLA generalization**.
> > >
> > > [1] Learning Dexterous In-Hand Manipulation. arXiv:1808.00177
> > >
> > > [2] Learning to Walk in Minutes Using Massively Parallel Deep Reinforcement Learning. CoRL 2022.
> > >
> > > [3] DeepMimic: Example-Guided Deep RL of Physics-Based Character Skills. SIGGRAPH 2018
> > >
> > > [4] Interactive Post-Training for Vision-Language-Action Models. arXiv:2505.17016
> > >
> > > [5] VLA-RL: Towards Masterful and General Robotic Manipulation with Scalable Reinforcement Learning. arXiv:2505.18719

---

### Official Review · Reviewer_QKnh · 2025-07-02

**Clarity:** 3
**Significance:** 3
**Originality:** 3
**Rating:** 5
**Confidence:** 4

**Summary:**

This paper presents an empirical evaluation of the effect of RL training of VLA models on their subsequent generalization performance.  Generalization is defined along Visual, Semantic, and Execution dimensions.  The paper also provides experiments and discussion to help those wishing to employ RL for fine-tuning choose the specific method to use.

**Questions:**

Some small points, discussion of which may improve the paper:

- In section 4.2, can you please expand on the choice of $h^0$?  What does the superscript there signify and what might the improved performance of that choice suggest about what information the critic is using.
- In line 280, the paper suggests that trial-and-error RL is able to generate a general "grasp" skill.  Why would this not work for vision to generate a general "perception" of graspable objects, as discussed immediately prior?  Can you clarify why those situations might be different?
- Could you address the motion-planning limitation in simple ways, by introducing noise, for example?

**Ethical Concerns:**

["NO or VERY MINOR ethics concerns only"]

**Final Justification:**

Thank you for your thorough rebuttal, which has brought some more clarity around the issues I raised.  I think this is a solid paper that should be accepted, and so will maintain my score.

**Limitations:**

yes

**Paper Formatting Concerns:**

None noticed

**Quality:**

3

**Strengths And Weaknesses:**

This paper is a very clear empirical study of a particular method and its effects on well-defined dimensions.  The problem tackled is well-motivated and timely, providing insight into tradeoffs for a difficult decision about how to fine-tune VLAs.  The scope of the paper is well defined, and the paper does not claim more than it shows.  The primary weaknesses of the paper are already identified by the authors:  motion-planner demonstrations on simple, single-task benchmarks.  Despite these weaknesses, the paper will be interesting to many who are interested in VLAs.

---

> ### Author Rebuttal · Authors · 2025-07-31
>
> We sincerely appreciate the reviewer’s supportive comments on our contributions and the potential impact on the community. Below, we address your concerns clearly and thoroughly.
>
> > ### Q1: Clarification on $h^0$ and analysis of critic information
>
> Thank you for pointing out the ambiguity regarding $h^0$. We will include an expanded and clarified explanation in the revised version.
> - **Detailed Explanation of $h^0$**: We adopt the commonly used discrete action tokenization approach from prior work [1,2], where each action dimension is discretized into 256 bins and subsequently tokenized, projecting actions into a token space. Specifically, we denote the final-layer embedding corresponding to each action token as $h^i$, where the index $i$ refers to the $i$-th action dimension. Consequently, $h^0$ specifically represents the embedding associated with the first action token.
> - **Rationale for Selecting $h^0$ as Critic Input**: In PPO, the critic is defined as the state-value function $V(s)$, which theoretically necessitates state-only information as input. In VLA models, observations encompass both visual inputs (images) and textual instructions, from which the VLM backbone extracts meaningful state representations. Then, a critical question arises: **Why specifically choose $h^0$ over other embeddings for approximating $V(s)$, i.e., $V(s)\approx V(h^0)$?**
>
>   Our hypothesis is that the embedding serving as input to the critic should predominantly capture state information with minimal interference from action-related information. Empirically, $h^0$ sits precisely at the boundary between state and action embeddings in the auto-regressive VLA. To substantiate this choice, we have conducted experiments (Fig. 4b) demonstrating significant performance degradation when using embeddings that encapsulate more action-oriented information—such as $h^n$, corresponding to the last action token, or $h^{all}$, representing the concatenation of all action token embeddings. We attribute this performance drop to action embeddings introducing unstable, noisy signals during training, inherently conflicting with PPO's requirement that critic inputs strictly represent state information.
>
> [1] OpenVLA: An Open-Source Vision-Language-Action Model. CoRL 2024.
>
> [2] RT-2: Vision-Language-Action Models Transfer Web Knowledge to Robotic Control. CoRL 2023.
>
> > ### Q2: Why RL works for "grasp" skills but not general "perception"
>
> We appreciate the reviewer’s thoughtful question. The distinction arises from which component of the VLA the generalization challenge primarily stresses. Our evaluation framework is explicitly designed to probe *Vision*, *Semantics*, and *Execution* generalization along separate axes, each of which targets different stages of the perception-to-action process.
>
> In **Semantics**, the challenge lies in generalizing across object identities, instructions, and receptacle configurations. RL benefits from trial-and-error by refining its policy through interaction—e.g., learning how to retry or adapt grasping behaviors in response to failures. As shown in Fig. 7 and Fig. 9, this enables RL to outperform SFT in unseen object scenarios by learning robust strategies.
>
> In contrast, **Vision** generalization introduces perturbations (e.g., novel textures, noise, unseen tables) that mostly challenge the perception module. While we do jointly train the vision encoder with RL, the signal from policy gradients remains sparse and task-driven, making it harder to recover from upstream perception errors such as mislocalization or occlusion. These errors degrade input quality before the agent can reason or act effectively. As a result, RL struggles to improve perception robustness beyond what is already achieved during pretraining, leading to comparable performance drops to SFT.
>
> We will clarify this in the revision to emphasize that all three axes—*Vision*, *Semantics*, and *Execution*—are essential and complementary in evaluating VLA generalization.
>
> > ### Q3: Limited motion-planning-based data collection.
>
> We appreciate the reviewer’s insightful comment regarding the use of motion-planner demonstrations. We acknowledge this as a potential limitation and have discussed it in the limitations section. In our experiments, we have tuned our motion planner to explore a larger space. Besides, we also tried to combine motion-planned data with demonstrations collected using a pretrained policy (e.g., Octo), as noted in Sec. 4.2 (lines 213–214). However, we found that the performance difference between motion-planned and Octo-collected data was minimal for SFT. Consequently, we scaled up motion-planning-based demonstrations (up to 64k) for our main SFT experiments.
>
> Regarding the suggestion of injecting noise into the planner, we note that while simple noise can increase variability, it also risks degrading demonstration quality for the whole dataset. Designing noise that balances meaningful exploration with high-quality trajectories is non-trivial. Alternatively, more principled approaches—such as integrating neural motion planners [3] or further employing hierarchical pipelines like ManipGen [4]—could offer more robust data diversity. These are promising directions we plan to explore, but given the time constraints of the review process, we leave them as future work.
>
> [3] Neural MP: A Generalist Neural Motion Planner. IROS 2025.
>
> [4] Local Policies Enable Zero-shot Long-Horizon Manipulation. ICRA 2025.

---

### Note · Authors · 2025-08-13

Dear AC and Reviewers,

Thank you for the thoughtful discussion. Our paper’s goal is a controlled, reproducible study of **what RL fine-tuning contributes to VLA generalization beyond SFT**.

**What stands after rebuttal**

- Across Vision / Semantics / Execution splits, RL improves **semantic understanding** and **execution robustness** over SFT while matching visual robustness.
- This conclusion persists with a stronger VLA (OpenVLA with action-chunking) and a more dexterous open-cabinet task with unseen articulations.
- We matched pipelines between SFT and RL, and released additional experiment details so the community can replicate and build upon this study.

**New evidence we added**

- **Preliminary real-robot validation**: a simulation-trained RL policy transfers better than SFT on a real-world Franka without policy retuning.
- **Architecture & task extensions**: the RL > SFT trend holds for stronger VLA model and in open-cabinet evaluation with perturbed poses/articulated objects.
- Expanded details: clarified visual perturbation tests, expanded details on data pipelines, critic inputs, advantage computation, and implementation for full reproducibility.

**Addressing remaining key concerns**

- **Why on-policy RL here**: In our simulation-controlled setting, on-policy RL is a principled way to optimize closed-loop behavior and reveals benefits SFT alone does not for multi-step control. Moreover, A broad range of influential robotics results and recent VLA studies have successfully leveraged on‑policy RL, supporting our design choice.
- **Beyond our current scope**: While broader RL variants warrant further study, we believe our current results already provide valuable insights and thorough benchmarks to the community, which are widely recognized by all reviewers. Preliminary real-robot validation also suggest that on-policy RL can improve real-world performance as simulation and sim-to-real pipelines mature.

**Final remarks**

RL fine-tuning brings **distinct, practically meaningful generalization gains**—especially in semantics and execution—beyond SFT, and these gains hold across models and tasks, with initial hardware validation pointing the same way. Several reviewers also characterized the setup as **systematic, careful, and timely**; we hope this summary helps the committee weigh the paper’s empirical contribution.

Sincerely,

The Authors

---

### Decision · Program_Chairs · 2025-09-17

**Decision:**

Accept (poster)

**Comment:**

This paper studies the generalization effects of VLAs trained with RL in comparison to SFT. They find that PPO outperforms both SFT and other RL algorithms (GRPO). The paper is practical with sufficient empirical analysis, is clearly written and has a well-supported story, and is generally useful to researchers wishing to fine-tune VLAs in robotic environments. The remaining concern of being limited to PPO / on-policy remains, but I disagree: on-policy algorithms are common in robotic settings and have been in use for RL with VLAs, e.g., https://arxiv.org/abs/2412.08442. It would be a nice bonus to see off-policy algorithm results, but I agree with the challenges and the limited impact on the story, so it isn't required.

I therefore recommend the paper is accepted as a poster.